# Poisson-Minibatching for Gibbs Sampling with Convergence Rate Guarantees

**Ruqi Zhang**
Cornell University
rz297@cornell.edu

**Christopher De Sa**
Cornell University
cdesa@cs.cornell.edu

## Abstract

Gibbs sampling is a Markov chain Monte Carlo method that is often used for learning and inference on graphical models. Minibatching, in which a small random subset of the graph is used at each iteration, can help make Gibbs sampling scale to large graphical models by reducing its computational cost. In this paper, we propose a new auxiliary-variable minibatched Gibbs sampling method, *Poisson-minibatching Gibbs*, which both produces unbiased samples and has a theoretical guarantee on its convergence rate. In comparison to previous minibatched Gibbs algorithms, Poisson-minibatching Gibbs supports fast sampling from continuous state spaces and avoids the need for a Metropolis-Hastings correction on discrete state spaces. We demonstrate the effectiveness of our method on multiple applications and in comparison with both plain Gibbs and previous minibatched methods.

## 1 Introduction

Gibbs sampling is a Markov chain Monte Carlo (MCMC) method which is widely used for inference on graphical models [7]. Gibbs sampling works by iteratively resampling a variable from its conditional distribution with the remaining variables fixed. Although Gibbs sampling is a powerful method, its utility can be limited by its computational cost when the model is large. One way to address this is to use *stochastic methods*, which use a subsample of the dataset or model—called a minibatch—to approximate the dataset or model used in an MCMC algorithm. Minibatched variants of many classical MCMC algorithms have been explored [18, 10, 3, 9], including the MIN-Gibbs algorithm for Gibbs sampling [3].

In this paper, we propose a new minibatched variant of Gibbs sampling on factor graphs called *Poisson-minibatching Gibbs* (Poisson-Gibbs). Like other minibatched MCMC methods, Poisson-minibatching Gibbs improves Gibbs sampling by reducing its computational cost. In comparison to prior work, our method improves upon MIN-Gibbs in two ways. First, it eliminates the need for a potentially expensive Metropolis-Hastings (M-H) acceptance step, giving it a better asymptotic per-iteration time complexity than MIN-Gibbs. Poisson-minibatching Gibbs is able to do this by choosing a minibatch in a way that depends on the current state of the variables, rather than choosing one that is independent of the current state as is usually done in stochastic algorithms. We show that such state-dependent minibatches can still be sampled quickly, and that an appropriately chosen state-dependent minibatch can result in a reversible Markov chain with the correct stationary distribution even without a Metropolis-Hastings correction step.

The second way that our method improves upon previous work is that it supports sampling over continuous state spaces, which are common in machine learning applications (in comparison, the previous work only supported sampling over discrete state spaces). The main difficulty here for Gibbs sampling is that resampling a continuous-valued variable from its conditional distribution requires sampling from a continuous distribution, and this is a nontrivial task (as compared with a discrete

| State Space | Algorithm | Computational Cost/Iter |
|---|---|---|
| Discrete | Gibbs sampling | $O(D\Delta)$ |
| | MIN-Gibbs [3] | $O(D\Psi^2)$ |
| | MGPMH [3] | $O(DL^2 + \Delta)$ |
| | DoubleMIN-Gibbs [3] | $O(DL^2 + \Psi^2)$ |
| | Poisson-Gibbs | $O(DL^2)$ |
| Continuous | Gibbs with rejection sampling | $O(N\Delta)$ |
| | PGITS: Poisson-Gibbs with ITS | $O(L^3)$ |
| | PGDA: Poisson-Gibbs with double approximation | $O(L^2 \log L)$ |

Table 1: Computational complexity cost for a single-iteration of Gibbs sampling. Here, $N$ is the required number of steps in rejection sampling to accept a sample, and the rest of the parameters are defined in Section 1.1.

random variable, which can be sampled from by explicitly computing its probability mass function). Our approach is based on fast inverse transform sampling method, which works by approximating the probability density function (PDF) of a distribution with a polynomial [13].

In addition to these two new capabilities, we prove bounds on the convergence rate of Poisson-minibatching Gibbs in comparison to plain (i.e. not minibatched) Gibbs sampling. These bounds can provide a recipe for how to set the minibatch size in order to come close to the convergence rate of plain Gibbs sampling. If we set the minibatch size in this way, we can derive expressions for the per-iteration computational cost of our method compared with others; these bounds are summarized in Table 1. In summary, the contributions of this paper are as follows:

- We introduce Poisson-minibatching Gibbs, a variant of Gibbs sampling which can reduce computational cost without adding bias or needing a Metropolis-Hastings correction step.

- We extend our method to sample from continuous-valued distributions.

- We prove bounds on the convergence rate of our algorithm, as measured by the spectral gap, on both discrete and continuous state spaces.

- We evaluate Poisson-minibatching Gibbs empirically, and show that its performance can match that of plain Gibbs sampling while using less computation at each iteration.

## 1.1 Preliminaries and Definitions

In this section, we present some background about Gibbs sampling and graphical models and give the definitions which will be used throughout the paper. In this paper, we consider Gibbs sampling on a factor graph [7], a type of graphical model that defines a probability distribution in terms of its *factors*. Explicitly, a factor graph consists of a set of variables $\mathcal{V}$ (each of which can take on values in some set $\mathcal{X}$) and a set of factors $\Phi$, and it defines a probability distribution $\pi$ over a state space $\Omega = \mathcal{X}^{\mathcal{V}}$, where the probability of some $x \in \Omega$ is

$$\pi(x) = \tfrac{1}{Z} \cdot \exp\left(\sum_{\phi \in \Phi} \phi(x)\right) = \tfrac{1}{Z} \cdot \prod_{\phi \in \Phi} \exp\left(\phi(x)\right).$$

Here, $Z$ denotes the scalar factor necessary for $\pi$ to be a distribution. Equivalently, we can think of this as the *Gibbs measure* with energy function

$$U(x) = \sum_{\phi \in \Phi} \phi(x), \qquad \text{where} \qquad \pi(x) \propto \exp(U(x));$$

this formulation will prove to be useful in many of the derivations later in the paper. (Here, the $\propto$ notation denotes that the expression on the left is a distribution that is proportional to the expression on the right with the appropriate constant of proportionality to make it a distribution.) In a factor graph, the factors $\phi$ typically only depend on a subset of the variables; we can represent this as a bipartite graph where the nodesets are $\mathcal{V}$ and $\Phi$ and where we draw an edge between a variable $i \in \mathcal{V}$ and a factor $\phi \in \Phi$ if $\phi$ depends on $i$. For simplicity, in this paper we assume that the variables are indexed with natural numbers $\mathcal{V} = \{1, \ldots, n\}$. We denote the set of factors that depend on the $i$th variable, as

$$A[i] = \{\phi | \phi \text{ depends on variable } i, \ \phi \in \Phi\}.$$

An important property of a factor graph is that the conditional distribution of a variable can be computed using only the factors that depend on that variable. This lends to a particularly efficient implementation of Gibbs sampling, in which only these adjacent factors are used at each iteration (rather than needing to evaluate the whole energy function $U$): this is illustrated in Algorithm 1.

The performance of our algorithm will depend on several parameters of the graphical model, which we will now restate, from previous work on MIN-Gibbs [3]. If the variables take on discrete values, we let $D = |\mathcal{X}|$ denote the number of values each can take on. We let $\Delta = \max_i |A[i]|$ denote the maximum degree of the graph. We assume that the magnitudes of the factor functions are all bounded, and for any $\phi$ we let $M_\phi$ denote this bound

---

**Algorithm 1** Gibbs Sampling

> **Input:** initial point $x$
> **loop**
>     **sample** variable $i \sim \mathrm{Unif}\{1, \ldots, n\}$
>     **for all** $v \in \mathcal{X}$ **do**
>         $x(i) \leftarrow v$
>         $U_v \leftarrow \sum_{\phi \in A[i]} \phi(x)$
>     **end for**
>     **construct distribution** $\rho$ where
>
> $$\rho(v) \propto \exp(U_v)$$
>
>     **sample** $v$ from $\rho$
>     **update** $x(i) \leftarrow v$
>     **output sample** $x$
> **end loop**

---

$$M_\phi = \left(\sup_{x \in \Omega} \phi(x)\right) - \left(\inf_{x \in \Omega} \phi(x)\right).$$

Without loss of generality (and as was done in previous works [3]), we will assume that $0 \le \phi(x) \le M_\phi$ because we can always add a constant to any factor $\phi$ without changing the distribution $\pi$. We define the *local maximum energy* $L$ and *total maximum energy* $\Psi$ of the graph as bounds on the sum of $M_\phi$ over the set of the factors associated with a single variable $i$ and the whole graph, respectively,

$$L = \max_{i \in \{1,2,\ldots,N\}} \sum_{\phi \in A[i]} M_\phi \qquad \text{and} \qquad \Psi = \sum_{\phi \in \Phi} M_\phi.$$

If the graph is very large and has many low-energy factors, the maximum energy of a graph can be much smaller than the maximum degree of the graph. All runtime analyses in this paper assume that evaluating a factor $\phi$ and sampling from a small discrete distribution can be done in constant time.

## 2 Poisson-Minibatching Gibbs Sampling

In this section, we will introduce the idea of Poisson-minibatching under the setting in which we assume we can sample from the conditional distribution of $x(i)$ exactly. One such example is when the state space of $x$ is discrete. We will consider how to sample from the conditional distribution when exact sampling is impossible in the next section.

In plain Gibbs sampling, we have to compute the sum over all the factors in $A[i]$ to get the energy in every step. When the graph is large, the computation of getting the energy can be expensive; for example, in the discrete case this cost is proportional to $D\Delta$. The main idea of Poisson-minibatching is to augment a desired distribution with extra Poisson random variables, which control how and whether a factor is used in the minibatch for a particular iteration. Maclaurin and Adams [10] used a similar idea to control whether a data point will be included in the minibatch or not with augmented Bernoulli variables. However, this method has been shown to be very inefficient when only updating a small fraction of Bernoulli variables in each iteration [15]. Our method does not suffer from the same issue due to the usage of Poisson variables which we will explain further later in this section.

We define the conditional distribution of additional variable $s_\phi$ for each factor $\phi$ as

$$s_\phi | x \sim \mathrm{Poisson}\left(\frac{\lambda M_\phi}{L} + \phi(x)\right)$$

where $\lambda > 0$ is a hyperparameter that controls the minibatch size. Then the joint distribution of variables $x$ and $s$, where $s$ is a variable vector including all $s_\phi$, is $\pi(x, s) = \pi(x) \cdot \mathbf{P}(s|x)$ and so

$$\pi(x, s) \propto \exp\left(\sum_{\phi \in \Phi}\left(s_\phi \log\left(1 + \frac{L}{\lambda M_\phi}\phi(x)\right) + s_\phi \log\left(\frac{\lambda M_\phi}{L}\right) - \log\left(s_\phi!\right)\right)\right). \quad (1)$$

Using (1) allows us to compute conditional distributions (of the variables $x_i$) using only a subset of the factors. This is because the factor $\phi$ will not contribute to the energy unless $s_\phi$ is greater than zero. If many $s_\phi$ are zero, then we only need to compute the energy over a small set of factors. Since

$$\mathbf{E}\left[|\{\phi \in A[i] \mid s_\phi > 0\}|\right] \le \mathbf{E}\left[\sum_{\phi \in A[i]} s_\phi\right] = \sum_{\phi \in A[i]}\left(\frac{\lambda M_\phi}{L} + \phi(x)\right) \le \lambda + L,$$

this implies that $\lambda + L$ is an upper bound of the expected number of non-zero $s_\phi$. When the graph is very large and has many low-energy factors, $\lambda + L$ can be much smaller than the factor set size, in which case only a small set of factors will contribute to the energy while most factor terms will disappear because $s_\phi$ is zero.

Using Poisson auxiliary variables has two benefits. First, compared with the Bernoulli auxiliary variables as described in FlyMC [10], there is a simple method for sampling $n$ Poisson random variables in total expected time proportional to the sum of their parameters, which can be much smaller than $n$ [3]. This means that sampling $n$ Poisson variables can be much more efficient than sampling $n$ Bernoulli variables, which allows our method to avoid any inefficiencies caused by sampling Bernoulli variables as in FlyMC. Second, compared with a fixed-minibatch-size method such as the one used in [18], Poisson-minibatching has the important property that the variables $s_\phi$ are independent. Whether a factor will be contained in the minibatch is independent to each other. This property is necessary for proving convergence rate theorems in the paper.

In Poisson-Gibbs, we will sample from the joint distribution alternately. At each iteration we can (1) first re-sample all the $s_\phi$, then (2) choose a variable index $i$ and re-sample $x(i)$. Here, we can reduce the state back to only $x$, since the future distribution never depends on the current value of $s$. Essentially, we only bother to re-sample the $s_\phi$ on which our eventual re-sampling of $x(i)$ depends: statistically, this is equivalent to re-sampling all $s_\phi$. Doing this corresponds to Algorithm 2.

However, minibatching by itself does not mean that the method must be more effective than plain Gibbs sampling. It is possible that the convergence rate of the minibatched chain becomes much slower than the original rate, such that the total cost of the minibatch method is larger than that of the baseline method even if the cost of each step is smaller. To rule out this undesirable situation, we prove that the convergence speed of our chain is not slowed down, or at least not too much, after applying minibatching. To do this, we bound the convergence rate of our algorithm, as measured by the *spectral gap* [8], which is the gap between the largest and second-largest eigenvalues of the chain's transition operator. This gap has been used previously to measure the convergence rate of minibatched MCMC [3].

**Theorem 1.** *Poisson-Gibbs (Algorithm 2) is reversible and has a stationary distribution $\pi$. Let $\bar{\gamma}$ denote its spectral gap, and let $\gamma$ denote the spectral gap of plain Gibbs sampling. If we use a minibatch size parameter $\lambda \geq 2L$, then*

$$\bar{\gamma} \geq \exp\left(-\frac{4L^2}{\lambda}\right) \cdot \gamma.$$

This theorem guarantees that the convergence rate of Poisson-Gibbs will not be slowed down by more than a factor of $\exp(-4L^2/\lambda)$. If we set $\lambda = \Theta(L^2)$, then this factor becomes $O(1)$, which is independent of the size of the problem. We proved Theorem 1 and the other theorems in this paper using the technique of Dirichlet forms, which is a standard way of comparing the spectral gaps of two chains by comparing their transition probabilities (more details are in the supplemental material).

Next, we derive expressions for the overall computational cost of Algorithm 2, supposing that we set $\lambda = \Theta(L^2)$ as suggested by Theorem 1. First, we need to evaluate the cost of sampling all the Poisson-distributed $s_\phi$. While a naïve approach to sample this would take $O(\Delta)$ time, we can do it substantially faster. For brevity, and because much of the technique is already described in the previous work [3], we defer an explicit analysis to the supplementary material, and just state the following.

**Statement 1.** *Sampling all the auxiliary variables $s_\phi$ for $\phi \in A[i]$ can be done in average time $O(\lambda + L)$, resulting in a sparse vector $s_\phi$.*

Now, to get an overall cost when assuming exact sampling from the conditional distribution, we consider discrete state spaces, in which we can sample from the conditional distribution of $x(i)$ exactly. In this case, the cost of a single iteration of Poisson-Gibbs will be dominated by the loop over $v$. This loop will run $D$ times, and each iteration will take $O(|S|)$ time to run. On average, this gives us an overall runtime $O((\lambda + L) \cdot D) = O(L^2 D)$ for Poisson-Gibbs. Note that due to the fast way we sample Poisson variables, the cost of sampling Poisson variables is negligible compared to other costs.

In comparison, the cost of the previous algorithms MIN-Gibbs, MGPMH and DoubleMIN-Gibbs [3] are all larger in big-$O$ than that of Poisson-Gibbs, as showed in Table 1. MGPMH and DoubleMIN-

Gibbs need to conduct an M-H correction, which adds to the cost, and the cost of MIN-Gibbs and DoubleMIN-Gibbs depend on $\Psi$ which is a global statistic. By contrast, our method does not need additional M-H step and is not dependent on global statistics. Thus the total cost of Gibbs sampling can be reduced more by Poisson-minibatching compared to the previous methods.

**Application of Poisson-Minibatching to Metropolis-Hastings.** Poisson-minibatching method can be applied to other MCMC methods, not just Gibbs sampling. To illustrate the general applicability of Poisson-minibatching method, we applied Poisson-minibatching to Metropolis-Hastings sampling and call it *Poisson-MH* (details of this algorithm and a demonstration on a mixture of Gaussians are given in the supplemental material). We get the following convergence rate bound.

**Theorem 2.** *Poisson-MH is reversible and has a stationary distribution $\pi$. If we let $\bar{\gamma}$ denote its spectral gap, and let $\bar{\gamma}$ denote the spectral gap of plain M-H sampling with the same proposal and target distributions, then*

$$\bar{\gamma} \geq \tfrac{1}{2} \exp\left(-\tfrac{L^2}{\lambda+L}\right) \cdot \gamma.$$

## 3 Poisson-Gibbs on Continuous State Spaces

In this section, we consider how to sample from a continuous conditional distribution, i.e. when $\mathcal{X} = [a, b] \subset \mathbb{R}$, without sacrificing the benefits of Poisson-minibatching. The main difficulty is that sampling from an arbitrary continuous conditional distribution is not trivial in the same way as sampling from an arbitrary discrete conditional distribution is. Some additional sampling method is required. In principle, we can combine any sampling method with Poisson-minibatching, such as rejection sampling which is commonly used in Gibbs sampling. However, rejection sampling needs to evaluate the energy multiple times per sample, so even if we reduce the cost of evaluating the energy by minibatching, the total cost can still be large, besides which there is no good guarantee on the convergence rate of rejection sampling.

In order to sample from the conditional distribution efficiently, we propose a new sampling method based on inverse transform sampling (ITS) method. The main idea is to approximate the continuous distribution with a polynomial; this requires only a number of energy function evaluations proportional to the degree of the polynomial. We provide overall cost and theoretical analysis of convergence rate for our method.

**Poisson-Gibbs with Double Chebyshev Approximation.** Inverse transform sampling is a classical method that generates samples from a uniform distribution and then transforms them by the inverse of cumulative distribution function (CDF) of the desired distribution. Since the CDF is often intractable in practice, Fast Inverse Transform Sampling (FITS) [13] uses a Chebyshev polynomial approximation to estimate the PDF fast and then get the CDF by computing an integral of a polynomial. Inspired by FITS, we propose Poisson-Gibbs with double Chebyshev approximation (PGDA).

The main idea of double Chebyshev approximation is to approximate the energy function first and then the PDF by using Chebyshev approximation *twice*. Specifically, we first get a polynomial approximation to the energy function $U$ on $[a, b]$, denoted by $\tilde{U}$, the *Chebyshev interpolant* [17]

$$\tilde{U}(x) = \sum_{k=0}^{m} \alpha_k T_k\left(\frac{2(x-a)}{b-a} - 1\right), \ \alpha_k \in \mathbb{R}, \ x \in [a, b], \tag{2}$$

where $T_k(x) = \cos(k \cos^{-1} x)$ is the degree-$k$ Chebyshev polynomial. Although the domain is continuous, we only need to evaluate $U$ on $m + 1$ Chebyshev nodes to construct the interpolant, and the expansion coefficients $\alpha_k$ can be computed stably in $O(m \log m)$ time. The following theorem shows that the error of a Chebyshev approximation can be made arbitrarily small with large $m$. (Although stated for the case of $[a, b] = [-1, 1]$, it easily generalizes to arbitrary $[a, b]$.)

**Theorem 3** (Theorem 8.2 from Trefethen [17]). *Assume $U$ is analytic in the open Bernstein ellipse $B([-1, 1], \rho)$, where the Bernstein ellipse is a region in the complex plane bounded by an ellipse with foci at $\pm 1$ and semimajor-plus-semiminor axis length $\rho > 1$. If for all $x \in B([-1, 1], \rho)$, $|U(x)| \leq V$ for some constant $V > 0$, the error of the Chebyshev interpolant on $[-1, 1]$ is bounded by*

$$|\tilde{U}(x) - U(x)| \leq \delta_m \qquad where \qquad \delta_m = \frac{4V\rho^{-m}}{\rho - 1}.$$

**Algorithm 2** Poisson-Gibbs

> **given:** initial state $x \in \Omega$
> **loop**
>   **sample** variable $i \sim \text{Unif}\{1, \dots, n\}$.
>   **for all** $\phi$ **in** $A[i]$ **do**
>     **sample** $s_\phi \sim \text{Poisson}\left(\frac{\lambda M_\phi}{L} + \phi(x)\right)$
>   **end for**
>   $S \leftarrow \{\phi | s_\phi > 0\}$
>   **for all** $v \in \mathcal{X}$ **do**
>     $x(i) \leftarrow v$
>     $U_v \leftarrow \sum_{\phi \in S} s_\phi \log\left(1 + \frac{L}{\lambda M_\phi}\phi(x)\right)$
>   **end for**
>   **construct distribution** $\rho$ where
>   $$\rho(v) \propto \exp(U_v)$$
>   **sample** $v$ from $\rho$
>   **update** $x(i) \leftarrow v$
>   **output sample** $x$
> **end loop**

**Algorithm 3** PGDA: Poisson-Gibbs Double Chebyshev Approximation

> **given:** state $x \in \Omega$, degree $m$ and $k$, domain $[a, b]$
> **loop**
>   **set** $i$, $s_\phi$, $S$, and $U$ as in Algorithm 2.
>   **construct** degree-$m$ Chebyshev polynomial approximation of energy $U_v$ on $[a, b]$: $\tilde{U}_v$
>   **construct** degree-$k$ Chebyshev polynomial approximation: $\tilde{f}(v) \approx \exp(\tilde{U}_v)$
>   **compute** the CDF polynomial
>   $$\tilde{F}(v) = \left(\int_a^b \tilde{f}(y)\,dy\right)^{-1} \int_a^v \tilde{f}(y)\,dy$$
>   **sample** $u \sim \text{Unif}[0, 1]$.
>   **solve** root-finding problem for $v$: $\tilde{F}(v) = u$
>   ▷ Metropolis-Hastings correction:
>   $p \leftarrow \frac{\exp(U_v)\tilde{f}(x(i))}{\exp(U_{x(i)})\tilde{f}(v)}$
>   **with probability** $\min(1, p)$, set $x(i) \leftarrow v$
>   **output sample** $x$
> **end loop**

After getting the approximation of the energy, we can get the PDF by $\exp(\tilde{U})$. However, it is generally hard to get the CDF now since the integral of $\exp(\tilde{U})$ for polynomial $\tilde{U}$ is usually intractable. So, we use *another* Chebyshev approximation $\tilde{f}$ to estimate $\exp(\tilde{U})$. Constructing the second Chebyshev approximation requires no additional evaluations of energy functions; its total computational cost is $\tilde{O}(mk)$ because we need to evaluate a degree-$m$ polynomial $k$ times to compute the coefficients. After doing this, we are able to compute the CDF directly since it is the integral of a polynomial. With the CDF $\tilde{F}(x)$ in hand, inverse transform sampling is used to generate samples. First, a pseudo-random sample $u$ is generated from the uniform distribution on $[0, 1]$, and then we solve the following root-finding problem for $x$: $\tilde{F}(x) = u$. Since $\tilde{F}(x)$ is a polynomial, this root-finding problem can be solved by many standard methods. We use bisection method to ensure the robustness of the algorithm [13].

Importantly, the sample we get here is actually from an *approximation* of the CDF. To correct the error introduced by the polynomial approximation, we add a M-H correction as the final step to make sure the samples come from the target distribution. Our algorithm is given in Algorithm 3. As before, we prove a bound on PGDA in terms of the spectral gap, given the additional assumption that the factors $\phi$ are analytic.

**Theorem 4.** *PGDA (Algorithm 3) is reversible and has a stationary distribution $\pi$. Let $\bar{\gamma}$ denote its spectral gap, and let $\gamma$ denote the spectral gap of plain Gibbs sampling. Assume $\rho > 1$ is some constant such that every factor function $\phi$, treated as a function of any single variable $x(i)$, must be analytically continuable to the Bernstein ellipse with radius parameter $\rho$ shifted-and-scaled so that its foci are at $a$ and $b$, such that it satisfies $|\phi(z)| \le M_\phi$ anywhere in that ellipse. Then, if $\lambda \log(2) \ge 4L$, and if $m$ is set large enough that $4\rho^{-m/2} \le \sqrt{\rho} - 1$, then it will hold that*

$$\bar{\gamma} \ge \left(1 - 4\sqrt{F}\right)\exp\left(\frac{-4L^2}{\lambda}\right) \cdot \gamma, \quad \text{where} \quad F = \frac{4 \cdot \exp(8L) \cdot \rho^{-\frac{k}{2}}}{\sqrt{\rho} - 1} + \exp\left(\frac{16L \cdot \rho^{-\frac{m}{2}}}{\sqrt{\rho} - 1}\right) - 1.$$

Similar to Theorem 1, this theorem implies that the convergence rate of PGDA can be slowed down by at most a constant factor relative to plain Gibbs. If we set $m = \Theta(\log L)$, $k = \Theta(L)$ and $\lambda = \Theta(L^2)$, then the ratio of the spectral gaps will also be $O(1)$, which is independent of the problem parameters. Note that it is possible to combine FITS with Poisson-Gibbs directly (i.e. use only one polynomial approximation to estimate the PDF directly), and we call this method *Poisson-Gibbs with fast inverse*

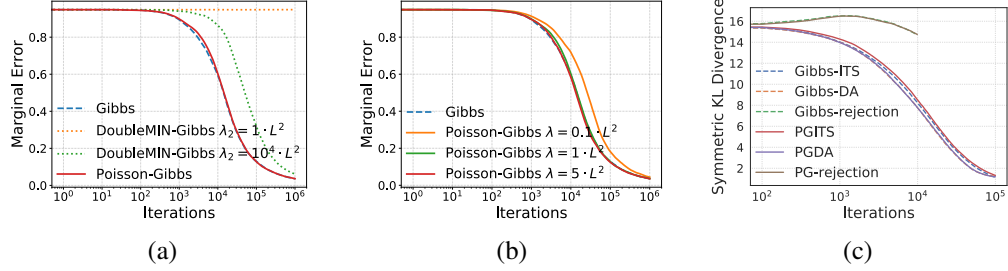

Figure 1: (a) Marginal error comparison among Poisson-Gibbs and previous methods on a Potts model. (b) Marginal error of Poisson-Gibbs on varying values of $\lambda$ on a Potts model. (c) Symmetric KL divergence comparison among PGITS, PGDA and previous methods on a continuous spin model.

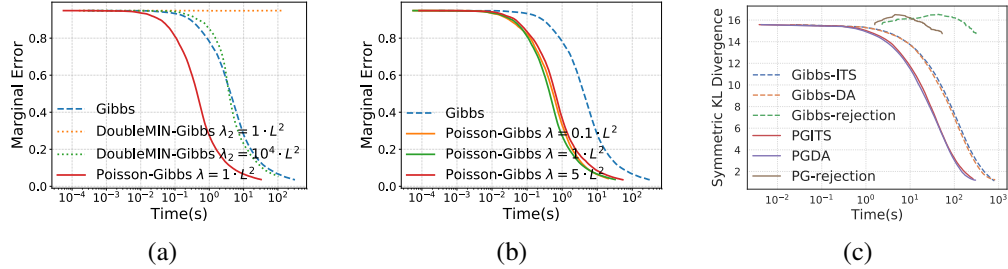

Figure 2: Runtime comparisons with the same experimental setting as in Figure 1.

*transform sampling* (PGITS). It turns out that PGDA is more efficient than PGITS since PGDA requires fewer evaluations of $U$ to achieve the same convergence rate. If we set the parameters as above, the total computational cost of PGDA is $O(m \cdot (\lambda + L) + m \cdot k) = O(\log L \cdot (L^2 + L)) = O(\log L \cdot L^2)$. On the other hand, the cost of PGITS to achieve the same constant-factor spectral gap ratio is $O(L^3)$. A derivation of this is given in the supplemental material.

## 4 Experiments

We demonstrate our methods on three tasks including Potts models, continuous spin models and truncated Gaussian mixture in comparison with plain Gibbs sampling and previous minibatched Gibbs sampling. We release the code at `https://github.com/ruqizhang/poisson-gibbs`.

### 4.1 Potts Models

We first test the performance of Poisson-minibatching Gibbs sampling on the Potts model [14] as in De Sa et al. [3]. The Potts model is a generalization of the Ising model [6] with domain $\{1, \ldots, D\}$ over an $N \times N$ lattice. The energy of a configuration is the following:

$$U(x) = \sum_{i=1}^{n} \sum_{j=1}^{n} \beta \cdot A_{ij} \cdot \delta\left(x(i), x(j)\right)$$

where the $\delta$ function equals one only when $x(i) = x(j)$ and zero otherwise. $A_{ij}$ is the interaction between two sites $i$ and $j$ and $\beta$ is the inverse temperature. As was done in previous work, we set the model to be fully connected and the interaction $A_{ij}$ is determined by the distance between site $i$ and site $j$ based on a Gaussian kernel [3]. The graph has $n = N^2 = 400$ variables in total, $\beta = 4.6$ and $D = 10$. On this model, $L = 5.09$.

We first compare our method with two other methods: plain Gibbs sampling and the most efficient MIN-Gibbs methods on this task, DoubleMIN-Gibbs. Note that, in comparison to our method, DoubleMIN-Gibbs needs an additional M-H correction step which requires a second minibatch to be sampled. We set $\lambda = 1 \cdot L^2$ for all minibatch methods. We tried two values for the second minibatch size in DoubleMIN-Gibbs $\lambda_2 = 1 \cdot L^2$ and $10^4 \cdot L^2$. We compute run-average marginal

distributions for each variable by collecting samples. By symmetry, the marginal for each variable in the stationary distribution is uniform, so the $\ell_2$-distance between the estimated marginals and the uniform distribution can be used to evaluate the convergence of Markov chain. We report this marginal error averaged over three runs.

Figure 1a shows the $\ell_2$-distance marginal error as a function of iterations. We observe that Poisson-Gibbs performs comparably with plain Gibbs and it outperforms DoubleMIN-Gibbs significantly especially when $\lambda_2$ is not large enough. The performance of DoubleMIN-Gibbs is highly influenced by the size of the second minibatch. We have to increase the second minibatch to $10^4 \cdot L^2$ in order to make it converge. This is because the variance of M-H correction will be very large when the second minibatch is not large enough. On the other hand, Poisson-Gibbs does not require an additional M-H correction which not only reduces the computational cost but also improves stability. In Figure 1b, we show the performance of our method with different values of $\lambda$. When we increase the minibatch size, the convergence speed of Poisson-Gibbs approaches plain Gibbs, which validates our theory. The number of factors being evaluated of Poisson-Gibbs varies each iteration, thus we report the average number which are 7, 28 and 132 respectively for $\lambda = 0.1 \cdot L^2$, $1 \cdot L^2$ and $5 \cdot L^2$.

The runtime comparisons with the same setup are reported in Figure 2a and 2b to demonstrate the computational speed-up of Poisson-Gibbs empirically. We can see that the results align with our theoretical analysis: Poisson-Gibbs is significantly faster than plain Gibbs samping and faster than previous minibatched Gibbs sampling methods. Compared to plain Gibbs, Poisson-Gibbs speeds up the computation by evaluating only a subset of factors in each iteration. Compared to DoubleMIN-Gibbs, Poisson-Gibbs is faster because it removes the need of an additional M-H correction step.

## 4.2 Continuous Spin Models

In this section, we study a more general setting of spin models where spins can take continuous values. Continuous spin models are of interest in both the statistics and physics communities [11, 2, 4]. This random graph model can also be used to describe complex networks such as social, information, and biological networks [12]. We consider the energy of a configuration as the following:

$$U(x) = \sum_{i=1}^{n} \sum_{j=1}^{n} \beta \cdot A_{ij} \cdot (x(i) \cdot x(j) + 1)$$

where $x(i) \in [0, 1]$ and $\beta = 1$. Notice that the existing minibatched Gibbs sampling methods [3] are not applicable on this task since they can be used only on discrete state spaces. We compare PGITS, PGDA with: (1) Gibbs sampling with FITS (Gibbs-ITS); (2) Gibbs sampling with Double Chebyshev approximation (Gibbs-DA); (3) Gibbs with rejection sampling (Gibbs-rejection); and (4) Poisson-Gibbs with rejection sampling (PG-rejection). We use symmetric KL divergence to quantitatively evaluate the convergence. On this model, $L = 13.71$ and we set $\lambda = L^2$. The degree of polynomial is $m = 3$ for PGITS and the first approximation in PGDA. The degree of polynomial is $k = 10$ for the second approximation in PGDA. In rejection sampling, we set the proposal distribution to be $wg$ where $g$ is the uniform distribution on $[0, 1]$ and $w$ is a constant tuned for best performance. The ground truth stationary distribution is obtained by running Gibbs-ITS for $10^7$ iterations.

On this task, the average number of evaluated factors per iteration of Poisson-Gibbs is 190. Figure 1c shows the symmetric KL divergence as a function of iterations, with results averaged over three runs. Observe that our methods achieve comparable performance to Gibbs sampling with only a fraction of factors. For rejection sampling, the average steps needed for a sample to be accepted is greater than 300 which means that the cost is much larger than that of PGITS and PGDA. Given the same time budget, it can only run for many fewer iterations (we run it for $10^4$ iterations). On the other hand, the two Chebyshebv approximation methods are much more efficient for both Poisson-Gibbs and plain Gibbs. The advantage of FITS over rejection sampling has also been discussed in previous work [13]. Also notice that PGDA converges faster than PGITS given the same degree of polynomial. This empirical result validates our theoretical results that suggest PGDA is more efficient than PGITS.

We also report the symmetric KL divergence as a function of runtime in Figure 2c. Similar to the previous section, the two Poisson-Gibbs methods are faster than plain Gibbs sampling.

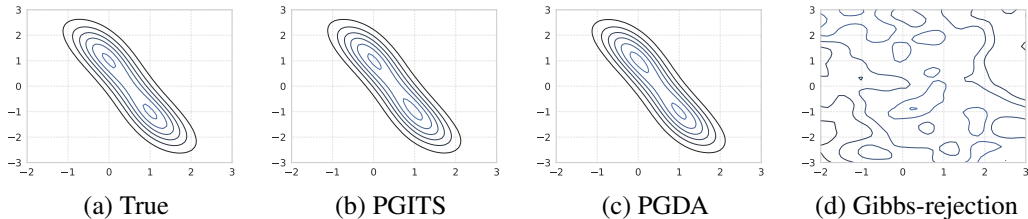

|     | (a) True | (b) PGITS | (c) PGDA | (d) Gibbs-rejection |

Figure 3: A visualization of the estimated density on a truncated Gaussian mixture model.

## 4.3 Truncated Gaussian Mixture

We further demonstrate PGITS and PGDA on a truncated Gaussian mixture model. We consider the following Gaussian mixture with tied means as done in previous work [18, 9]:

$$x_1 \sim \mathcal{N}(0, \sigma_1^2), \ x_2 \sim \mathcal{N}(0, \sigma_2^2), \ y_i \sim \frac{1}{2}\mathcal{N}(x_1, \sigma_y^2) + \frac{1}{2}\mathcal{N}(x_1 + x_2, \sigma_y^2).$$

We used the same parameters as in Welling and Teh [18]: $\sigma_1^2 = 10$, $\sigma_2^2 = 1$, $\sigma_y^2 = 2$, $x_1 = 0$ and $x_2 = 1$. This posterior has two modes at $(x_1, x_2) = (0, 1)$ and $(x_1, x_2) = (1, -1)$. We truncate the posterior by bounding the variables $x_1$ and $x_2$ in $[-6, 6]$. The energy can be written as

$$U(x) = \log p(x_1) + \log p(x_2) + \sum_{i=1}^{N} \log p(y_i | x_1, x_2)$$

which can be regarded as a factor graph with $N$ factors. We add a positive constant to the energy to ensure each factor is non-negative: this will not change the underlying distribution. As in Li and Wong [9], we set $N = 10^6$. $L = 1581.14$ for this model and we set $\lambda = 500$, $m = 20$ and $k = 25$. We have also considered higher values of $\lambda$ and found that the results are very similar. We generate $10^6$ samples for all methods. A uniform distribution in $[-6, 6]$ is used as the proposal distribution in Gibbs with rejection sampling. We try varying values for $w$ but none of them results in reasonable density estimate which may be due to the inefficiency of rejection sampling [13]. We report the results when the average needed steps for a sample to be accepted is around 1000. The average number of factors being evaluated per iteration of Poisson-Gibbs is 1802. Our results are reported in Figure 3, where we observe visually that the density estimates of PGITS and PGDA are very accurate. In contrast, rejection sampling completely failed to estimate the density given the budget.

## 5 Conclusion

We propose Poisson-minibatching Gibbs sampling to generate unbiased samples with theoretical guarantees on the convergence rate. Our method provably converges to the desired stationary distribution at a rate that is at most a constant factor slower than the full batch method, as measured by the spectral gap. We provide guidance about how to set the hyperparameters of our method to make the convergence speed arbitrarily close to the full batch method. On continuous state spaces, we propose two variants of Poisson-Gibbs based on fast inverse transform sampling and provide convergence analysis for both of them. We hope that our work will help inspire more exploration into unbiased and guaranteed-fast stochastic MCMC methods.

**Acknowledgements**

This work was supported by a gift from Huawei. We thank Wing Wong for the helpful discussion.

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
