[Supplementary Material]

# Supplementary Material: Poisson-Minibatching for Gibbs Sampling with Convergence Rate Guarantees

## A  Fast Sampling of the Auxiliary Variables

In this section, we describe in detail the method used to sample the auxiliary variables $s_\phi$ and prove Statement 1. The method for doing so is described here in Algorithm 4.

---
**Algorithm 4** Sample auxiliary variables $s_\phi$
---
  ▷ pre-computation step; happens once
  **for** $i = 1$ **to** $n$ **do**
     $\Lambda_i \leftarrow \sum_{\phi \in A[i]} \frac{\lambda M_\phi}{L} + M_\phi$
     **compute distribution** $\rho_i$ over $A[i]$ where

$$\rho_i(\phi) \propto \frac{\lambda M_\phi}{L} + M_\phi.$$

  **process distribution** $\rho_i$ so that in future, it can be sampled from in constant time
  **end for**

  ▷ to actually re-sample the auxiliary variables
  **given:** current state $x \in \Omega$, variable $i$ to resample
  **initialize sparse vector** $s : A[i] \to \mathbb{Z}$
  **sample** $B \sim \text{Poisson}(\Lambda_i)$
  **for** $b = 1$ **to** $B$ **do**
     **sample** $\phi \sim \rho_i$
     **compute** $\phi(x)$
     **with probability** $\frac{\frac{\lambda M_\phi}{L} + \phi(x)}{\frac{\lambda M_\phi}{L} + M_\phi}$ **update** sparse vector $s_\phi \leftarrow s_\phi + 1$
  **end for**

---

To see that this is valid, let $B = \sum_i^n s_i$ where $s_i$ are Poisson variables with parameters $\lambda_i$. We know that $B$ is also Poisson distributed with parameter $\Lambda = \sum_i^n \lambda_i$. Conditioned on the value of $B$, it is known that $s_i$ follows a multinomial distribution with event probabilities $\lambda_i/\Lambda$ and trial count $B$. Therefore, we can first sample $B \sim \text{Poisson}(\Lambda)$ and then sample

$$(s_1, \ldots s_n) \sim \text{Multinomial}\left( B, \left( \frac{\lambda_1}{\Lambda}, \ldots, \frac{\lambda_n}{\Lambda} \right) \right).$$

Our Algorithm 4 is only slightly more complicated than this process, in order to minimize the number of times that $\phi(x)$ is evaluated, but it can be seen to produce the valid distribution by the same reasoning.

The computational cost of Algorithm 4 is clearly proportional to $B$, and since

$$\mathbf{E}[B] = \Lambda_i = \sum_{\phi \in A[i]} \frac{\lambda M_\phi}{L} + M_\phi \le \lambda + L,$$

it follows that the overall average computational cost will also be $\lambda + L$. This proves Statement 1.

## B  Poisson-Gibbs with Exact Sampling from the Conditional Distribution

### B.1  Derivation of the joint distribution

In this subsection, we derive the joint distribution (1) by substituting the distributions of $x$ and $s$ into the conditional distribution of $s$ given $x$. By the expression of Poisson distribution for $s_\phi$ and the

independence of $s_\phi$, we have

$$\pi(x, s) = \pi(x)\pi(s|x)$$

$$\propto \exp\left(\sum_{\phi \in \Phi} \phi(x)\right) \prod_{\phi \in \Phi} \pi(s_\phi|x)$$

$$= \exp\left(\sum_{\phi \in \Phi} (\phi(x) + \log \pi(s_\phi|x))\right)$$

$$= \exp\left(\sum_{\phi \in \Phi} \left(\phi(x) + s_\phi \log\left(\frac{\lambda M_\phi}{L} + \phi(x)\right) - \log(s_\phi!) - \frac{\lambda M_\phi}{L} - \phi(x)\right)\right)$$

$$= \exp\left(\sum_{\phi \in \Phi} \left(s_\phi \log\left(\frac{\lambda M_\phi}{L} + \phi(x)\right) - \log(s_\phi!) - \frac{\lambda M_\phi}{L}\right)\right)$$

$$\propto \exp\left(\sum_{\phi \in \Phi} \left(s_\phi \log\left(\frac{\lambda M_\phi}{L} + \phi(x)\right) - \log(s_\phi!)\right)\right)$$

$$= \exp\left(\sum_{\phi \in \Phi} \left(s_\phi \log\left(1 + \frac{L}{\lambda M_\phi}\phi(x)\right) + s_\phi \log\left(\frac{\lambda M_\phi}{L}\right) - \log(s_\phi!)\right)\right).$$

## B.2 Proof of Theorem 1

In this section, we prove that Poisson-Gibbs converges, and derive a bound on its convergence rate.

*Proof.* First, we will derive an expression for the transition operator of Poisson-Gibbs chain, and show it is reversible. Then we will bound the spectral gap.

If $x$ and $y$ are states which differ in only one variable $i$, the probability of transitioning from $x$ to $y$ will be the probability of choosing to sample variable $i$ times the expected value over the random choice of $s$ of the probability of sampling $y(i)$ from $\rho$. That is,

$$T(x, y) = \frac{1}{n} \cdot \mathbf{E}\left[\rho(y(i))\right]$$

$$= \frac{1}{n} \cdot \mathbf{E}\left[\frac{\exp(U_{y(i)})}{\int \exp(U_u)\, du}\right]$$

$$= \frac{1}{n} \cdot \sum_s \frac{\exp(U_{y(i)})}{\int \exp(U_u)\, du} \cdot \prod_{\phi \in A[i]} \frac{1}{s_\phi!}\left(\frac{\lambda M_\phi}{L} + \phi(x)\right)^{s_\phi} \exp\left(-\left(\frac{\lambda M_\phi}{L} + \phi(x)\right)\right)$$

$$= \frac{1}{n} \cdot \sum_s \frac{\exp\left(\sum_{\phi \in A[i]} s_\phi \log\left(\frac{\lambda M_\phi}{L} + \phi(y)\right)\right)}{\int \exp\left(\sum_{\phi \in A[i]} s_\phi \log\left(\frac{\lambda M_\phi}{L} + \phi(z_u)\right)\right)\, du}$$

$$\cdot \prod_{\phi \in A[i]} \left(1 + \frac{L}{\lambda M_\phi}\phi(x)\right)^{s_\phi} \cdot \exp\left(-\phi(x)\right)$$

$$\cdot \prod_{\phi \in A[i]} \frac{1}{s_\phi!}\left(\frac{\lambda M_\phi}{L}\right)^{s_\phi} \cdot \exp\left(-\frac{\lambda M_\phi}{L}\right)$$

where $z_u$ denotes $x$ where $x(i)$ has been set equal to $u$. Note that $s_\phi$ here are non-negative integers that a Poisson variable can take, not variables. So if we let $r_\phi \sim \text{Poisson}\left(\frac{\lambda M_\phi}{L}\right)$ and $r_\phi$ to be all

independent, we can write this as

$$
\begin{aligned}
T(x,y) &= \frac{1}{n} \cdot \mathbf{E}_r \left[ \frac{\exp\left(\sum_{\phi \in A[i]} r_\phi \log\left(\frac{\lambda M_\phi}{L} + \phi(y)\right)\right)}{\int \exp\left(\sum_{\phi \in A[i]} r_\phi \log\left(\frac{\lambda M_\phi}{L} + \phi(z_u)\right)\right) \, du} \right. \\
&\qquad \left. \cdot \prod_{\phi \in A[i]} \left(1 + \frac{L}{\lambda M_\phi}\phi(x)\right)^{r_\phi} \cdot \exp\left(-\phi(x)\right) \right] \\
&= \frac{1}{n} \cdot \mathbf{E}_r \left[ \frac{\exp\left(\sum_{\phi \in A[i]} r_\phi \left(\log\left(1 + \frac{L}{\lambda M_\phi}\phi(y)\right) + \log\left(1 + \frac{L}{\lambda M_\phi}\phi(x)\right)\right)\right)}{\int \exp\left(\sum_{\phi \in A[i]} r_\phi \log\left(1 + \frac{L}{\lambda M_\phi}\phi(z_u)\right)\right) \, du} \right. \\
&\qquad \left. \cdot \exp\left(-\sum_{\phi \in A[i]} \phi(x)\right) \right]
\end{aligned}
$$

Therefore, since

$$
\pi(x) = \frac{1}{Z} \cdot \exp\left(\sum_{\phi \in \Phi} \phi(x)\right),
$$

it follows that

$$
\begin{aligned}
&\pi(x)T(x,y) \\
&= \frac{1}{nZ} \cdot \mathbf{E}_r \left[ \frac{\exp\left(\sum_{\phi \in A[i]} r_\phi \left(\log\left(1 + \frac{L}{\lambda M_\phi}\phi(y)\right) + \log\left(1 + \frac{L}{\lambda M_\phi}\phi(x)\right)\right)\right)}{\int \exp\left(\sum_{\phi \in A[i]} r_\phi \log\left(1 + \frac{L}{\lambda M_\phi}\phi(z_u)\right)\right) \, du} \right. \\
&\qquad \left. \cdot \exp\left(\sum_{\phi \in \Phi} \phi(x) - \sum_{\phi \in A[i]} \phi(x)\right) \right] \\
&= \frac{\exp(U_{\neg i}(x))}{nZ} \cdot \mathbf{E}_r \left[ \frac{\exp\left(\sum_{\phi \in A[i]} r_\phi \left(\log\left(1 + \frac{L}{\lambda M_\phi}\phi(y)\right) + \log\left(1 + \frac{L}{\lambda M_\phi}\phi(x)\right)\right)\right)}{\int \exp\left(\sum_{\phi \in A[i]} r_\phi \log\left(1 + \frac{L}{\lambda M_\phi}\phi(z_u)\right)\right) \, du} \right].
\end{aligned}
$$

where we define $U_{\neg i}(x) = \sum_{\phi \notin A[i]} \phi(x)$. This expression is symmetric in $x$ and $y$ (note that $U_{\neg i}(x)$ does not depend on variable $i$), so it follows that the Markov chain is reversible, and its stationary distribution is indeed $\pi$.

We can proceed to try to bound its spectral gap, using the technique of Dirichlet forms. We start by simplifying our expression by defining

$$
\bar{\phi}(x) = \frac{L\phi(x)}{\lambda M_\phi}.
$$

Using this, we get

$$
\pi(x)T(x,y) = \frac{\exp(U_{\neg i}(x))}{nZ} \cdot \mathbf{E}_r \left[ \frac{\exp\left(\sum_{\phi \in A[i]} r_\phi \left(\log\left(1 + \bar{\phi}(y)\right) + \log\left(1 + \bar{\phi}(x)\right)\right)\right)}{\int \exp\left(\sum_{\phi \in A[i]} r_\phi \log\left(1 + \bar{\phi}(z_u)\right)\right) \, du} \right].
$$

We proceed by bringing the exponential on the top of this sum down to the bottom and inside the integral, which produces

$$\pi(x)T(x,y) = \frac{\exp\left(U_{\neg i}(x)\right)}{nZ} \cdot \mathbf{E}_r\left[\left(\int \exp\left(\sum_{\phi \in A[i]} r_\phi\left(\log\left(1 + \bar{\phi}(z_u)\right)\right.\right.\right.\right.$$
$$\left.\left.\left.\left. - \log\left(1 + \bar{\phi}(x)\right) - \log\left(1 + \bar{\phi}(y)\right)\right)\right) du\right)^{-1}\right]$$
$$\geq \frac{\exp\left(U_{\neg i}(x)\right)}{nZ} \cdot \left(\mathbf{E}_r\left[\int \exp\left(\sum_{\phi \in A[i]} r_\phi\left(\log\left(1 + \bar{\phi}(z_u)\right)\right.\right.\right.\right.$$
$$\left.\left.\left.\left. - \log\left(1 + \bar{\phi}(x)\right) - \log\left(1 + \bar{\phi}(y)\right)\right)\right) du\right]\right)^{-1}$$

where this inequality follows from Jensen's inequality and the fact that $1/x$ is convex. By converting the exp-of-sum to a product-of-exp, and recalling that the $r_\phi$ are independent, we can further reduce this to

$$\pi(x)T(x,y) \geq \frac{\exp\left(U_{\neg i}(x)\right)}{nZ}\left(\int \mathbf{E}_r\left[\prod_{\phi \in A[i]} \exp\left(r_\phi\left(\log\left(1 + \bar{\phi}(z_u)\right)\right.\right.\right.\right.$$
$$\left.\left.\left.\left. - \log\left(1 + \bar{\phi}(x)\right) - \log\left(1 + \bar{\phi}(y)\right)\right)\right)\right] du\right)^{-1}$$
$$= \frac{\exp\left(U_{\neg i}(x)\right)}{nZ}\left(\int \prod_{\phi \in A[i]} \mathbf{E}_r\left[\exp\left(r_\phi\left(\log\left(1 + \bar{\phi}(z_u)\right)\right.\right.\right.\right.$$
$$\left.\left.\left.\left. - \log\left(1 + \bar{\phi}(x)\right) - \log\left(1 + \bar{\phi}(y)\right)\right)\right)\right] du\right)^{-1}.$$

This final expectation expression is just the moment generating function of the Poisson random variable $r_\phi$ evaluated at

$$t = \log\left(1 + \bar{\phi}(z_u)\right) - \log\left(1 + \bar{\phi}(x)\right) - \log\left(1 + \bar{\phi}(y)\right).$$

Here, from the standard formula for that MGF, we get

$$\mathbf{E}_r[\exp(r_\phi t)] = \exp\left(\frac{\lambda M_\phi}{L}\left(\exp(t) - 1\right)\right)$$

So

$$\exp(t) - 1$$
$$= \frac{1 + \bar{\phi}(z_u)}{(1 + \bar{\phi}(x))(1 + \bar{\phi}(y))} - 1$$
$$= \frac{\bar{\phi}(z_u) - \bar{\phi}(x) - \bar{\phi}(y) - \bar{\phi}(x)\bar{\phi}(y)}{(1 + \bar{\phi}(x))(1 + \bar{\phi}(y))}$$
$$= \bar{\phi}(z_u) - \bar{\phi}(x) - \bar{\phi}(y) - \frac{\left(\bar{\phi}(z_u) - \bar{\phi}(x) - \bar{\phi}(y)\right)\left(\bar{\phi}(x) + \bar{\phi}(y) + \bar{\phi}(x)\bar{\phi}(y)\right) + \bar{\phi}(x)\bar{\phi}(y)}{(1 + \bar{\phi}(x))(1 + \bar{\phi}(y))}.$$

Since

$$0 \leq \bar{\phi}(x) = \frac{L\phi(x)}{\lambda M_\phi} \leq \frac{L}{\lambda} \leq \frac{1}{2}$$

(where here we're using the condition in the theorem statement that $2L \leq \lambda$) we can bound this with

$$\exp(t) - 1$$

$$\leq \bar{\phi}(z_u) - \bar{\phi}(x) - \bar{\phi}(y) - \frac{\left(-\bar{\phi}(x) - \bar{\phi}(y)\right)\left(\bar{\phi}(x) + \bar{\phi}(y)\right) + \left(1 - \bar{\phi}(x) - \bar{\phi}(y)\right)\bar{\phi}(x)\bar{\phi}(y)}{(1 + \bar{\phi}(x))(1 + \bar{\phi}(y))}$$

$$\leq \bar{\phi}(z_u) - \bar{\phi}(x) - \bar{\phi}(y) + \frac{\left(\bar{\phi}(x) + \bar{\phi}(y)\right)\left(\bar{\phi}(x) + \bar{\phi}(y)\right)}{(1 + \bar{\phi}(x))(1 + \bar{\phi}(y))}$$

$$\leq \bar{\phi}(z_u) - \bar{\phi}(x) - \bar{\phi}(y) + \left(\bar{\phi}(x) + \bar{\phi}(y)\right)^2$$

$$\leq \bar{\phi}(z_u) - \bar{\phi}(x) - \bar{\phi}(y) + \frac{4L^2}{\lambda^2}.$$

So,

$$\mathbf{E}\exp(r_\phi t) = \exp\left(\frac{\lambda M_\phi}{L}\left(\exp(t) - 1\right)\right)$$

$$\leq \exp\left(\frac{\lambda M_\phi}{L}\left(\bar{\phi}(z_u) - \bar{\phi}(x) - \bar{\phi}(y) + \frac{4L^2}{\lambda^2}\right)\right)$$

$$= \exp\left(\phi(z_u) - \phi(x) - \phi(y) + \frac{4LM_\phi}{\lambda}\right).$$

Substituting this into the original expression produces

$$\pi(x)T(x, y)$$

$$\geq \frac{\exp\left(U_{\neg i}(x)\right)}{nZ}\left(\int \prod_{\phi \in A[i]} \exp\left(\phi(z_u) - \phi(x) - \phi(y) + \frac{4LM_\phi}{\lambda}\right) du\right)^{-1}$$

$$= \frac{\exp\left(U_{\neg i}(x)\right)}{nZ}\left(\int \exp\left(\sum_{\phi \in A[i]} \phi(z_u) - \sum_{\phi \in A[i]} \phi(x) - \sum_{\phi \in A[i]} \phi(y) + \sum_{\phi \in A[i]} \frac{4LM_\phi}{\lambda}\right) du\right)^{-1}$$

$$\geq \frac{\exp\left(U_{\neg i}(x)\right)}{nZ}\left(\int \exp\left(\sum_{\phi \in A[i]} \phi(z_u) - \sum_{\phi \in A[i]} \phi(x) - \sum_{\phi \in A[i]} \phi(y) + \frac{4L^2}{\lambda}\right) du\right)^{-1}$$

$$= \exp\left(-\frac{4L^2}{\lambda}\right)\frac{\exp\left(U_{\neg i}(x)\right)}{nZ}\left(\int \exp\left(\bar{U}_u - \bar{U}_{x(i)} - \bar{U}_{y(i)}\right) du\right)^{-1}$$

$$= \exp\left(-\frac{4L^2}{\lambda}\right)\frac{\exp\left(U_{\neg i}(x)\right)}{nZ}\frac{\exp(\bar{U}_{x(i)}) \cdot \exp(\bar{U}_{y(i)})}{\int \exp(\bar{U}_u)du}$$

$$= \exp\left(-\frac{4L^2}{\lambda}\right)\frac{1}{nZ}\frac{\exp(U(x)) \cdot \exp(\bar{U}_{y(i)})}{\int \exp(\bar{U}_u)du}$$

where $\bar{U}_v$ denotes the assignment of $U_v$ in the plain Gibbs sampling algorithm (Algorithm 1),

$$\bar{U}_v = \sum_{\phi \in A[i]} \phi(z_v),$$

Finally, if we let $G$ denote the transition probability operator of plain Gibbs sampling, we notice right away that

$$\pi(x)T(x, y) \geq \exp\left(-\frac{4L^2}{\lambda}\right)\frac{1}{nZ}\frac{\exp(U(x)) \cdot \exp(\bar{U}_{y(i)})}{\int \exp(\bar{U}_u)du}$$

$$= \exp\left(-\frac{4L^2}{\lambda}\right)\pi(x)G(x, y).$$

We will use the Dirichlet form argument to finish the proof. A real function $f$ is square integrable with respect to probability measure $\pi$, if it satisfies

$$\int f(x)^2 \pi(dx) < \infty.$$

Define $L^2(\pi)$ to be the Hilbert space of all such functions.

Let $L_0^2(\pi) \subset L^2(\pi)$ to be the Hilbert space that uses the same inner product but only contains functions such that

$$\mathbf{E}_\pi[f] = \int f(x)\pi(dx) = 0.$$

We also define the notation

$$\langle f, g \rangle = \int f(x)g(x)\pi(dx).$$

A special example is $\mathrm{Var}_\pi[f] = \langle f, f \rangle$.

From here, the Dirichlet form of a Markov chain associated with transition operator $T$ is given by [5]

$$\mathbf{E}(f) = \frac{1}{2} \int \int (f(x) - f(y))^2 T(x,y)\pi(x)dxdy.$$

And the spectral gap can be written as [1]

$$\gamma = \inf_{f \in L_0^2(\pi):\mathrm{Var}_\pi[f]=1} \mathbf{E}(f).$$

The spectral gap is related to other common measurement of the convergence of MCMC. For example, it has the following relationship with the mean squared error $e_\pi$ on a Markov chain $\{X_n\}_{n \in \mathbb{N}}$ [16],

$$e_\pi^2 \leq \frac{2}{n\gamma} \|f\|_2^2.$$

With the expression of the spectral gap, it follows that

$$
\begin{aligned}
\bar{\gamma} &= \inf_{f \in L_0^2(\pi):Var_\pi[f]=1} \left[ \frac{1}{2} \int \int (f(x) - f(y))^2 T(x,y)\pi(x) \, dx \, dy \right] \\
&\geq \exp\left(-\frac{4L^2}{\lambda}\right) \cdot \inf_{f \in L_0^2(\pi):Var_\pi[f]=1} \left[ \frac{1}{2} \int \int (f(x) - f(y))^2 G(x,y)\pi(x) \, dx \, dy \right] \\
&= \exp\left(-\frac{4L^2}{\lambda}\right) \cdot \gamma.
\end{aligned}
$$

This proves the theorem. $\qquad\square$

## C  Poisson-Gibbs on Continuous State Spaces

### C.1  Poisson-Gibbs with Fast Inverse Transform Sampling (PGITS)

In the main body of the paper, we mentioned the PGITS method, Poisson-Gibbs with Fast Inverse Transform Sampling. This method is to approximate the PDF by Chebyshev polynomials and then use inverse transform sampling. In this section, we will outline the algorithm and derive convergence rate results for it. These results will illustrate why PGITS can be expected to perform worse than PGDA.

PGITS operates by approximating the PDF with a Chebyshev polynomial approximation and then sampling from that polynomial approximation using inverse transform sampling. Specifically, if the PDF we want to sample from is $f(x)$, we can approximate $f$ by $\tilde{f}$ on $[a,b]$ using Chebyshev polynomials,

$$\tilde{f} = \sum_{k=0}^{m} \alpha_k T_k \left( \frac{2(x-a)}{b-a} - 1 \right), \ \alpha_k \in \mathbb{R}, \ x \in [a,b] \tag{3}$$

where $T_k(x) = \cos(k \cos^{-1} x)$ is the degree $k$ Chebyshev polynomial, and $\alpha_k$ are the Chebyshev coefficients of the function $f$ [17]. We do this by interpolating $f$ at its Chebyshev nodes, resulting in $\tilde{f}$ being the $m$th order *Chebyshev interpolant*. Once we have the polynomial approximation $\tilde{f}$ we

---

**Algorithm 5** PGITS: Poisson-Gibbs Inverse Transform Sampling

---

**given:** state $x \in \Omega$, degree $m$, domain $[a, b]$
**loop**
    **set** $i$, $s_\phi$, $S$, and $U$ as in Algorithm 2.
    **construct** degree-$m$ Chebyshev polynomial approximation of polynomial PDF on $[a, b]$

$$\tilde{f}(v) \approx \exp(U_v)$$

    **compute** the CDF polynomial

$$\tilde{F}(v) = \left( \int_a^b \tilde{f}(y)\ dy \right)^{-1} \int_a^v \tilde{f}(y)\ dy$$

    **sample** $u \sim \mathrm{Unif}[0, 1]$.
    **solve** root-finding problem for $v$: $\tilde{F}(v) = u$
    ▷ Metropolis-Hastings correction:

$$p \leftarrow \frac{\exp(U_v) \cdot \tilde{f}(x(i))}{\exp(U_{x(i)}) \cdot \tilde{f}(v)}$$

    **with probability** $\min(1, p)$, set $x(i) \leftarrow v$
    **output sample** $x$
**end loop**

---

can construct the corresponding CDF approximation $\tilde{F}$ by calculating the integral directly (since polynomials are straightforward to integrate). With the approximation $\tilde{F}$, we are able to use inverse transform sampling to generate samples. We call this whole algorithm *PGITS* and it is listed as Algorithm 5.

We show that PGITS is reversible and bound its spectral gap in the following theorem.

**Theorem 5.** *PGITS (Algorithm 5) is reversible and has a stationary distribution $\pi$. Let $\bar{\gamma}$ denote its spectral gap, and let $\gamma$ denote the spectral gap of plain Gibbs sampling. Assume $\rho > 1$ is some constant such that every factor function $\phi$, treated as a function of any single variable $x(i)$, must be analytically continuable to the Bernstein ellipse with radius parameter $\rho$ shifted-and-scaled so that its foci are at $a$ and $b$, such that it satisfies $|\phi(z)| \le M_\phi$ anywhere in that ellipse. Then, if $\lambda \ge 2L$ it will hold that*

$$\bar{\gamma} \ge \left( 1 - \frac{8 \exp(L) \rho^{-m/2}}{\sqrt{\rho - 1}} \right) \cdot \exp\left( -\frac{4L^2}{\lambda} \right) \cdot \gamma.$$

We can set $m = \Theta(L)$ and $\lambda = \Theta(L^2)$ to make the ratio of the spectral gaps $O(1)$, which is independent of the size of the problem. If the parameters are set in this way, the total cost of PGITS is $O(m \cdot (\lambda + L)) = O(L \cdot L^2) = O(L^3)$.

### C.1.1 Proof of Theorem 5

*Proof.* Similar to the previous analysis of Poisson-Gibbs, we will show the PGITS is reversible by using the expression of the transition operator. Then we will bound the spectral gap.

Let $T_{i,s}(x, y)$ denote the probability of transitioning from state $x$ to $y$ given that we have already chosen to sample variable $i$ with minibatch coefficients $s$. Then, the overall transition operator will be

$$T(x, y) = \mathbf{E}\left[ T_{i,s}(x, y) \right]$$

where the expectation is taken over $i$ and $s$.

Let the polynomial interpolant for $\exp(U_v)$ be $\tilde{f}(v)$ which is given in (3). Note that this interpolant is a function of the index $i$ and the minibatch coefficients $s$. Then,

$$T_{i,s}(x, y) = \rho(y(i)) \cdot \min(1, a)$$

$$= \frac{\tilde{f}(y(i))}{\int \tilde{f}(u)du} \cdot \min\left(1, \frac{\exp(U_{y(i)})\tilde{f}(x(i))}{\exp(U_{x(i)})\tilde{f}(y(i))}\right)$$

Therefore,

$$T(x,y) = \frac{1}{n}\mathbf{E}\frac{\tilde{f}(y(i))}{\int \tilde{f}(u)du} \cdot \min\left(1, \frac{\exp(U_{y(i)})\tilde{f}(x(i))}{\exp(U_{x(i)})\tilde{f}(y(i))}\right)$$

$$= \frac{1}{n}\mathbf{E}\frac{1}{\int \tilde{f}(u)du} \cdot \min\left(\tilde{f}(y(i)), \exp(U_{y(i)} - U_{x(i)})\tilde{f}(x(i))\right)$$

$$= \frac{1}{n}\mathbf{E}\frac{1}{\int \tilde{f}(u)du} \cdot \min\left(\tilde{f}(y(i)), \exp\left(\sum_{\phi \in A[i]} s_\phi \log \frac{1 + \frac{L}{\lambda M_\phi}\phi(y)}{1 + \frac{L}{\lambda M_\phi}\phi(x)}\right)\tilde{f}(x(i))\right)$$

$$= \frac{1}{n}\sum_s \frac{1}{\int \tilde{f}(u)du} \cdot \min\left(\tilde{f}(y(i)), \exp\left(\sum_{\phi \in A[i]} s_\phi \log \frac{1 + \frac{L}{\lambda M_\phi}\phi(y)}{1 + \frac{L}{\lambda M_\phi}\phi(x)}\right)\tilde{f}(x(i))\right)$$

$$\cdot \exp\left(\sum_{\phi \in A[i]} s_\phi \log\left(\frac{\lambda M_\phi}{L} + \phi(x)\right) - \log\left(s_\phi!\right) - \left(\frac{\lambda M_\phi}{L} + \phi(x)\right)\right)$$

$$= \frac{1}{n}\sum_s \frac{1}{\int \tilde{f}(u)du} \cdot \min\left(\tilde{f}(y(i))\exp\left(\sum_{\phi \in A[i]} s_\phi \log\left(1 + \frac{L}{\lambda M_\phi}\phi(x)\right)\right),\right.$$

$$\exp\left(\sum_{\phi \in A[i]} s_\phi \log\left(1 + \frac{L}{\lambda M_\phi}\phi(y)\right)\right)\tilde{f}(x(i))\right)$$

$$\left.\cdot \exp\left(\sum_{\phi \in A[i]}\left(s_\phi \log\left(\frac{\lambda M_\phi}{L}\right) - \log\left(s_\phi!\right) - \left(\frac{\lambda M_\phi}{L} + \phi(x)\right)\right)\right)\right)$$

$$= \frac{1}{n}\sum_s \frac{1}{\int \tilde{f}(u)du} \cdot \min\left(\tilde{f}(y(i))\exp\left(\sum_{\phi \in A[i]} s_\phi \log\left(1 + \frac{L}{\lambda M_\phi}\phi(x)\right)\right),\right.$$

$$\exp\left(\sum_{\phi \in A[i]} s_\phi \log\left(1 + \frac{L}{\lambda M_\phi}\phi(y)\right)\right)\tilde{f}(x(i))\right)$$

$$\left.\cdot \exp\left(\sum_{\phi \in A[i]}\left(s_\phi \log\left(\frac{\lambda M_\phi}{L}\right) - \log\left(s_\phi!\right) - \left(\frac{\lambda M_\phi}{L}\right)\right)\right)\right)\exp(-U_{x(i)})$$

Multiplying $\pi(x)$ on both sides,

$$\pi(x)T(x,y)$$

$$= \frac{\exp(U_{\neg i}(x))}{nZ}\sum_s \frac{1}{\int \tilde{f}(u)du} \cdot \min\left(\tilde{f}(y(i))\exp\left(\sum_{\phi \in A[i]} s_\phi \log\left(1 + \frac{L}{\lambda M_\phi}\phi(x)\right)\right),\right.$$

$$\tilde{f}(x(i))\exp\left(\sum_{\phi \in A[i]} s_\phi \log\left(1 + \frac{L}{\lambda M_\phi}\phi(y)\right)\right)\right)$$

$$\left.\cdot \exp\left(\sum_{\phi \in A[i]}\left(s_\phi \log\left(\frac{\lambda M_\phi}{L}\right) - \log\left(s_\phi!\right) - \left(\frac{\lambda M_\phi}{L}\right)\right)\right)\right)$$

This expression is symmetric in $x$ and $y$, so it follows that

$$\pi(x)T(x,y) = \pi(y)T(y,x)$$

Thus the Markov chain is reversible, and its stationary distribution is $\pi$.

We now bound its spectral gap, using the technique of Dirichlet forms. First, as before, we start by re-writing the chain in terms of an expectation of a new random variable $r_\phi$ where $r_\phi \sim \text{Poisson}\left(\frac{\lambda M_\phi}{L}\right)$ and the $r_\phi$ are all independent. We also define $\bar{\phi}(x) = \frac{L\phi(x)}{\lambda M_\phi}$ as before. This gives us

$$\pi(x)T(x,y) = \frac{\exp(U_{\neg i}(x))}{nZ} \mathbf{E}_r\left[\frac{1}{\int \tilde{f}(u)du} \cdot \min\left(\tilde{f}(y(i))\exp\left(\sum_{\phi \in A[i]} r_\phi \log\left(1 + \bar{\phi}(x)\right)\right),\right.\right.$$

$$\left.\left.\tilde{f}(x(i))\exp\left(\sum_{\phi \in A[i]} r_\phi \log\left(1 + \bar{\phi}(y)\right)\right)\right)\right]$$

$$= \frac{\exp(U_{\neg i}(x))}{nZ}\mathbf{E}_r\left[\frac{1}{\int \tilde{f}(u)du} \cdot \min\left(\tilde{f}(y(i))\exp\left(U_{x(i)}\right), \tilde{f}(x(i))\exp\left(U_{y(i)}\right)\right)\right]$$

where now the $\tilde{f}$ are considered to be a function of $r_\phi$ rather than $s_\phi$ as before.

To proceed further we will need to use the fact that $\tilde{f}$ is a Chebyshev interpolant to bound its error compared with $U$. Recall that, here,

$$U_v = \sum_{\phi \in A[i]} r_\phi \log\left(1 + \frac{L}{\lambda M_\phi}\phi(z_v)\right) = \sum_{\phi \in A[i]} r_\phi \log\left(1 + \bar{\phi}(z_v)\right),$$

and $\tilde{f}(v) \approx \exp(U_v)$ in the sense of being a degree-$m$ Chebyshev polynomial interpolant. Recall that we assumed that the each function $\phi$, treated as a function in any single variable, must be analytic on a (shifted) Bernstein ellipse on the interval $[a,b]$ with parameter $\rho$ (i.e. a standard Bernstein ellipse on $[-1,1]$ with parameter $\rho$ shifted and scaled to have its foci at $a$ and $b$), and that its magnitude must be bounded by

$$|\phi(z)| \le M_\phi$$

for any $z$ in this ellipse (keeping all the other parameters as usual within $[a,b]$. It follows that the magnitude of the function $U_v$ is bounded by

$$|\exp(U_v)| = \left|\exp\left(\sum_{\phi \in A[i]} r_\phi \log\left(1 + \frac{L}{\lambda M_\phi}\phi(z_v)\right)\right)\right|$$

$$= \prod_{\phi \in A[i]}\left|1 + \frac{L}{\lambda M_\phi}\phi(z_v)\right|^{r_\phi}$$

$$\le \prod_{\phi \in A[i]}\left(1 + \frac{L}{\lambda}\right)^{r_\phi}.$$

Therefore, from Theorem 3, we know that

$$\left|\tilde{f}(v) - \exp(U_v)\right| \le \frac{4\rho^{-m}}{\rho - 1} \cdot \prod_{\phi \in A[i]}\left(1 + \frac{L}{\lambda}\right)^{r_\phi} = \frac{4\rho^{-m}}{\rho - 1} \cdot \left(1 + \frac{L}{\lambda}\right)^{\sum_{\phi \in A[i]} r_\phi}.$$

Since we also assumed that $\phi(z)$ is always non-negative, $U_v$ must also be non-negative, and so in particular $\exp(-U_v) \le 1$, so

$$\left|\frac{\tilde{f}(v)}{\exp(U_v)} - 1\right| \le \frac{4\rho^{-m}}{\rho - 1} \cdot \left(1 + \frac{L}{\lambda}\right)^{\sum_{\phi \in A[i]} r_\phi} \le \frac{4\rho^{-m}}{\rho - 1} \cdot \exp\left(\frac{L}{\lambda}\sum_{\phi \in A[i]} r_\phi\right).$$

If we now define

$$C = \frac{4\rho^{-m}}{\rho - 1} \cdot \exp\left(\frac{L}{\lambda}\sum_{\phi \in A[i]} r_\phi\right),$$

then
$$(1 - C) \cdot \exp(U_v) \leq \tilde{f}(v) \leq (1 + C) \cdot \exp(U_v).$$

In particular, this means that
$$\min\left(\tilde{f}(y(i)) \exp\left(U_{x(i)}\right), \tilde{f}(x(i)) \exp\left(U_{y(i)}\right)\right) \geq (1 - C) \cdot \exp\left(U_{x(i)} + U_{y(i)}\right),$$

and
$$\frac{1}{\int \tilde{f}(u) \, du} \geq \frac{1}{1 + C} \cdot \frac{1}{\int \exp(U_u) \, du}.$$

Substituting this into our bound above gives
$$\pi(x)T(x,y) \geq \frac{\exp(U_{\neg i}(x))}{nZ} \, \mathbf{E}_r \left[\frac{1 - C}{1 + C} \cdot \frac{\exp\left(U_{x(i)} + U_{y(i)}\right)}{\int \exp(U_u) \, du}\right].$$

Now, recall that we set this up by sampling $r_\phi$ independently from a Poisson random variable $r_\phi \sim \text{Poisson}\left(\frac{\lambda M_\phi}{L}\right)$. This distribution is equivalent to assigning
$$\Lambda = \sum_{\phi \in A[i]} \frac{\lambda M_\phi}{L},$$

sampling the random variable $B \sim \text{Poisson}(\Lambda)$, and then sampling $r_\phi \sim \text{Multinomial}\left(B, \frac{\lambda M_\phi}{\Lambda L}\right)$. If we re-think our distribution as coming from this process, then by the Law of Total Expectation,
$$\pi(x)T(x,y) \geq \frac{\exp(U_{\neg i}(x))}{nZ} \, \mathbf{E}_B \left[\frac{1 - C}{1 + C} \cdot \mathbf{E}_r \left[\frac{\exp\left(U_{x(i)} + U_{y(i)}\right)}{\int \exp(U_u) \, du}\bigg| B\right]\right],$$

where we can pull out the terms in $C$ because we can write $C$ to depend only on $B$ as
$$C = \frac{4\rho^{-m}}{\rho - 1} \cdot \exp\left(\frac{L}{\lambda} \sum_{\phi \in A[i]} r_\phi\right) = \frac{4\rho^{-m}}{\rho - 1} \cdot \exp\left(\frac{LB}{\lambda}\right).$$

Next, we can bound this inner expectation with
$$\mathbf{E}_r \left[\frac{\exp\left(U_{x(i)} + U_{y(i)}\right)}{\int \exp(U_u) \, du}\bigg| B\right]$$
$$= \mathbf{E}_r \left[\frac{1}{\int \exp(U_u - U_{x(i)} - U_{y(i)}) \, du}\bigg| B\right]$$
$$\geq \mathbf{E}_r \left[\int \exp(U_u - U_{x(i)} - U_{y(i)}) \, du\bigg| B\right]^{-1}$$
$$= \mathbf{E}_r \left[\int \exp\left(\sum_{\phi \in A[i]} r_\phi \left(\log\left(1 + \bar{\phi}(z_u)\right) - \log\left(1 + \bar{\phi}(x)\right) - \log\left(1 + \bar{\phi}(y)\right)\right)\right) \, du\bigg| B\right]^{-1}$$
$$= \mathbf{E}_r \left[\int \exp\left(\sum_{\phi \in A[i]} r_\phi t_\phi\right) \, du\bigg| B\right]^{-1}$$
$$= \left(\int \mathbf{E}_r \left[\exp\left(\sum_{\phi \in A[i]} r_\phi t_\phi\right)\bigg| B\right] \, du\right)^{-1},$$

where we define
$$t_\phi = \log\left(1 + \bar{\phi}(z_u)\right) - \log\left(1 + \bar{\phi}(x)\right) - \log\left(1 + \bar{\phi}(y)\right).$$

This inner expectation is now just the moment-generating function of the multinomial distribution. Applying the standard formula for that MGF gives us

$$\mathbf{E}_r\left[\exp\left(\sum_{\phi\in A[i]} r_\phi t_\phi\right)\bigg| B\right] = \left(\sum_{\phi\in A[i]} \frac{\lambda M_\phi}{\Lambda L}\cdot\exp(t_\phi)\right)^B.$$

Substituting this back into our original expression gives

$$\pi(x)T(x,y) \geq \frac{\exp(U_{\neg i}(x))}{nZ}\ \mathbf{E}_B\left[\frac{1-C}{1+C}\cdot\left(\int\left(\sum_{\phi\in A[i]} \frac{\lambda M_\phi}{\Lambda L}\cdot\exp(t_\phi)\right)^B du\right)^{-1}\right].$$

Next, let $\delta > 0$ be a small constant, to be assigned later. Recall that for any non-negative random variable $X$ and any event $A$, by the Law of Total Probability,

$$\mathbf{E}[X] = \mathbf{E}[X|A]\cdot\mathbf{P}(A) + \mathbf{E}[X|\neg A]\cdot\mathbf{P}(\neg A) \geq \mathbf{E}[X|A]\cdot\mathbf{P}(A).$$

So, since the interior of this expectation is a non-negative number, it follows that

$$\pi(x)T(x,y) \geq \frac{\exp(U_{\neg i}(x))}{nZ}\ \mathbf{E}_B\left[\frac{1-C}{1+C}\cdot\left(\int\left(\sum_{\phi\in A[i]} \frac{\lambda M_\phi}{\Lambda L}\cdot\exp(t_\phi)\right)^B du\right)^{-1}\bigg| C\leq\delta\right]$$
$$\cdot\mathbf{P}_B(C\leq\delta)$$
$$\geq \frac{\exp(U_{\neg i}(x))}{nZ}\cdot\frac{1-\delta}{1+\delta}\cdot\mathbf{E}_B\left[\left(\int\left(\sum_{\phi\in A[i]} \frac{\lambda M_\phi}{\Lambda L}\cdot\exp(t_\phi)\right)^B du\right)^{-1}\bigg| C\leq\delta\right]$$
$$\cdot\mathbf{P}_B(C\leq\delta).$$

By Jensen's inequality again, we get

$$\pi(x)T(x,y) \geq \frac{\exp(U_{\neg i}(x))}{nZ}\ \mathbf{E}_B\left[\frac{1-C}{1+C}\cdot\left(\int\left(\sum_{\phi\in A[i]} \frac{\lambda M_\phi}{\Lambda L}\cdot\exp(t_\phi)\right)^B du\right)^{-1}\bigg| C\leq\delta\right]$$
$$\cdot\mathbf{P}_B(C\leq\delta)$$
$$\geq \frac{\exp(U_{\neg i}(x))}{nZ}\cdot\frac{1-\delta}{1+\delta}\cdot\left(\int\mathbf{E}_B\left[\left(\sum_{\phi\in A[i]} \frac{\lambda M_\phi}{\Lambda L}\cdot\exp(t_\phi)\right)^B\bigg| C\leq\delta\right] du\right)^{-1}$$
$$\cdot\mathbf{P}_B(C\leq\delta).$$

Since this inner expectation is again non-negative, we can again apply our above inequality, but in the opposite direction, giving

$$\mathbf{E}[X|A] \leq \frac{\mathbf{E}[X]}{\mathbf{P}(A)}.$$

This produces

$$\pi(x)T(x,y) \geq \frac{\exp(U_{\neg i}(x))}{nZ}\cdot\frac{1-\delta}{1+\delta}\cdot\left(\int\mathbf{E}_B\left[\left(\sum_{\phi\in A[i]} \frac{\lambda M_\phi}{\Lambda L}\cdot\exp(t_\phi)\right)^B\right] du\right)^{-1}$$
$$\cdot\mathbf{P}_B(C\leq\delta)^2.$$

Now, we are just left with the MGF of a Poisson-distributed random variable. This we already know to be

$$
\mathbf{E}_B\left[\left(\sum_{\phi \in A[i]} \frac{\lambda M_\phi}{\Lambda L} \cdot \exp(t_\phi)\right)^B\right] = \mathbf{E}_B\left[\exp\left(B \log\left(\sum_{\phi \in A[i]} \frac{\lambda M_\phi}{\Lambda L} \cdot \exp(t_\phi)\right)\right)\right]
$$

$$
= \exp\left(\Lambda\left(\left(\sum_{\phi \in A[i]} \frac{\lambda M_\phi}{\Lambda L} \cdot \exp(t_\phi)\right) - 1\right)\right)
$$

$$
= \exp\left(\sum_{\phi \in A[i]} \frac{\lambda M_\phi}{L} \cdot (\exp(t_\phi) - 1)\right),
$$

where in the last line we can leverage the fact that

$$
\sum_{\phi \in A[i]} \frac{\lambda M_\phi}{\Lambda L} = 1
$$

to justify pulling the $-1$ inside the sum. From the analysis of Poisson-Gibbs, we had that

$$
\exp(t_\phi) - 1 \leq \bar{\phi}(z_u) - \bar{\phi}(x) - \bar{\phi}(y) + \frac{4L^2}{\lambda^2}.
$$

So,

$$
\mathbf{E}_B\left[\left(\sum_{\phi \in A[i]} \frac{\lambda M_\phi}{\Lambda L} \cdot \exp(t_\phi)\right)^B\right] \leq \exp\left(\sum_{\phi \in A[i]} \frac{\lambda M_\phi}{L} \cdot \left(\bar{\phi}(z_u) - \bar{\phi}(x) - \bar{\phi}(y) + \frac{4L^2}{\lambda^2}\right)\right)
$$

$$
= \exp\left(\sum_{\phi \in A[i]} \left(\phi(z_u) - \phi(x) - \phi(y) + \frac{4LM_\phi}{\lambda}\right)\right)
$$

$$
\leq \exp\left(\bar{U}_u - \bar{U}_{x(i)} - \bar{U}_{y(i)} + \frac{4L^2}{\lambda}\right),
$$

where as in the analysis of Poisson-Gibbs, $\bar{U}_v$ denotes the assignment of $U_v$ in the plain Gibbs sampling algorithm (Algorithm 1),

$$
\bar{U}_v = \sum_{\phi \in A[i]} \phi(z_v).
$$

Substituting this expression in to our overall bound, we get

$$
\pi(x)T(x,y) \geq \frac{\exp(U_{\neg i}(x))}{nZ} \cdot \frac{1-\delta}{1+\delta} \cdot \left(\int \exp\left(\bar{U}_u - \bar{U}_{x(i)} - \bar{U}_{y(i)} + \frac{4L^2}{\lambda}\right) du\right)^{-1}
$$

$$
\cdot \mathbf{P}_B(C \leq \delta)^2
$$

$$
= \frac{\exp(U(x))}{nZ} \cdot \frac{1-\delta}{1+\delta} \cdot \frac{\exp(\bar{U}_{y(i)})}{\int \exp(\bar{U}_u) \, du}
$$

$$
\cdot \exp\left(-\frac{4L^2}{\lambda}\right) \cdot \mathbf{P}_B(C \leq \delta)^2.
$$

Finally, if we let $G$ denote the transition probability operator of plain Gibbs sampling, we notice right away that

$$
\pi(x)T(x,y) \geq \frac{1-\delta}{1+\delta} \cdot \exp\left(-\frac{4L^2}{\lambda}\right) \cdot \mathbf{P}_B(C \leq \delta)^2 \cdot \pi(x)G(x,y)
$$

$$
\geq (1-2\delta) \cdot \exp\left(-\frac{4L^2}{\lambda}\right) \cdot \mathbf{P}_B(C \leq \delta)^2 \cdot \pi(x)G(x,y).
$$

To get a final bound, all we need to do is bound $\mathbf{P}_B(C \leq \delta)$. This is straightforward, since

$$\mathbf{P}_B(C \leq \delta) = \mathbf{P}_B\left(\frac{4\rho^{-m}}{\rho - 1} \cdot \exp\left(\frac{LB}{\lambda}\right) \leq \delta\right)$$

$$= \mathbf{P}_B\left(\exp\left(\frac{LB}{\lambda}\right) \leq \frac{\rho - 1}{4\rho^{-m}} \cdot \delta\right).$$

Notice that by the MGF formula for $B$,

$$\mathbf{E}_B\left[\exp\left(\frac{LB}{\lambda}\right)\right] \leq \exp\left(\Lambda\left(\exp\left(\frac{L}{\lambda}\right) - 1\right)\right).$$

Since we chose a minibatch size parameter $\lambda \geq 2L$, it follows that $L/\lambda \leq 1/2$, and so

$$\exp\left(\frac{L}{\lambda}\right) - 1 \leq \frac{2L}{\lambda},$$

and so since also

$$\Lambda = \sum_{\phi \in A[i]} \frac{\lambda M_\phi}{L} \leq \lambda.$$

it follows that

$$\mathbf{E}_B\left[\exp\left(\frac{LB}{\lambda}\right)\right] \leq \exp\left(\lambda \cdot \frac{2L}{\lambda}\right) = \exp(2L).$$

Therefore, by Markov's inequality,

$$\mathbf{P}_B(C \geq \delta) = \mathbf{P}_B\left(\exp\left(\frac{LB}{\lambda}\right) \geq \frac{\rho - 1}{4\rho^{-m}} \cdot \delta\right)$$

$$\leq \frac{\exp(2L)}{\frac{\rho - 1}{4\rho^{-m}} \cdot \delta}$$

$$\leq \frac{4\rho^{-m}}{\rho - 1} \cdot \frac{\exp(2L)}{\delta}.$$

Thus,

$$\mathbf{P}_B(C \leq \delta) = 1 - \mathbf{P}_B(C \geq \delta)$$

$$\geq 1 - \frac{4\rho^{-m}}{\rho - 1} \cdot \frac{\exp(2L)}{\delta},$$

and in particular

$$\mathbf{P}_B(C \leq \delta)^2 = (1 - \mathbf{P}_B(C \geq \delta))^2$$

$$\geq 1 - 2\mathbf{P}_B(C \geq \delta)$$

$$\geq 1 - \frac{8\rho^{-m}}{\rho - 1} \cdot \frac{\exp(2L)}{\delta}.$$

Substituting this back into our overall bound gives us

$$\pi(x)T(x,y) \geq \frac{1 - \delta}{1 + \delta} \cdot \exp\left(-\frac{4L^2}{\lambda}\right) \cdot \mathbf{P}_B(C \leq \delta)^2 \cdot \pi(x)G(x,y)$$

$$\geq (1 - 2\delta) \cdot \left(1 - \frac{8\rho^{-m}}{\rho - 1} \cdot \frac{\exp(2L)}{\delta}\right) \cdot \exp\left(-\frac{4L^2}{\lambda}\right) \cdot \pi(x)G(x,y)$$

$$\geq \left(1 - 2\delta - \frac{8\rho^{-m}}{\rho - 1} \cdot \frac{\exp(2L)}{\delta}\right) \cdot \exp\left(-\frac{4L^2}{\lambda}\right) \cdot \pi(x)G(x,y).$$

Finally, choosing the value of $\delta$ as

$$\delta = \frac{2\exp(L)}{\rho^{m/2} \cdot \sqrt{\rho - 1}},$$

we get

$$\pi(x)T(x,y) \geq \left(1 - \frac{8\exp(L)\rho^{-m/2}}{\sqrt{\rho-1}}\right) \cdot \exp\left(-\frac{4L^2}{\lambda}\right) \cdot \pi(x)G(x,y).$$

Now applying the standard Dirichlet form argument, we get

$$\bar{\gamma} \geq \left(1 - \frac{8\exp(L)\rho^{-m/2}}{\sqrt{\rho-1}}\right) \cdot \exp\left(-\frac{4L^2}{\lambda}\right) \cdot \gamma,$$

which was the desired expression. $\qquad\square$

## C.2  Proof of Theorem 4

*Proof.* The reversibility can be proved by the same procedure as in Section C.1.1. By applying that same analysis, which did not depend on the manner in which the approximation $\tilde{f}$ was constructed, we can arrive at the expression

$$\pi(x)T(x,y) = \frac{\exp(U_{\neg i}(x))}{nZ} \, \mathbf{E}_r \left[\frac{1}{\int \tilde{f}(u)du} \cdot \min\left(\tilde{f}(y(i))\exp\left(U_{x(i)}\right), \tilde{f}(x(i))\exp\left(U_{y(i)}\right)\right)\right].$$

By the assumption of $\phi(z)$, we have

$$
\begin{aligned}
|U_v| &= \left|\sum_{\phi \in A[i]} r_\phi \log\left(1 + \bar{\phi}(z_v)\right)\right| \\
&\leq \sum_{\phi \in A[i]} r_\phi \left|\log\left(1 + \frac{L}{\lambda M_\phi}\phi(x)\right)\right| \\
&\leq \sum_{\phi \in A[i]} r_\phi \left|\frac{2L}{\lambda M_\phi}\phi(x)\right| \\
&\leq \frac{2L}{\lambda} \sum_{\phi \in A[i]} r_\phi.
\end{aligned}
$$

where the second inequality holds because

$$|z| \leq \frac{1}{2} \quad \Rightarrow \quad |\log(1+z)| \leq 2\,|z|,$$

using the assumptions $\lambda \geq 2L$ and $|\phi(x)| \leq M_\phi$. Now applying Lemma 1 in Section E, assigning $\sigma = \sqrt{\rho}$ gives us,

$$\left|\tilde{U}_v - U_v\right| \leq \frac{8\rho^{-\frac{m}{2}}}{\sqrt{\rho}-1} \cdot \frac{L}{\lambda} \sum_{\phi \in A[i]} r_\phi,$$

for any $v$ in the shifted-and-scaled Bernstein ellipse with parameter $\sqrt{\rho}$.

Next, since $\tilde{U}_v$ is a polynomial in $v$, $\exp(\tilde{U}_v)$ must be analytic everywhere in $\mathbb{C}$. In particular it must be analytic on the Bernstein ellipse on the interval $[a,b]$ with parameter $\sqrt{\rho}$. On that interval, it is bounded by

$$
\begin{aligned}
\left|\exp(\tilde{U}_v)\right| &\leq \exp\left(\left|\tilde{U}_v\right|\right) \\
&\leq \exp\left(|U_v| + \left|\tilde{U}_v - U_v\right|\right) \\
&\leq \exp\left(\frac{2L}{\lambda}\sum_{\phi \in A[i]} r_\phi\right) \cdot \exp\left(\frac{8\rho^{-\frac{m}{2}}}{\sqrt{\rho}-1} \cdot \frac{L}{\lambda}\sum_{\phi \in A[i]} r_\phi\right) \\
&\leq \exp\left(\frac{4\rho^{-\frac{m}{2}} + \sqrt{\rho}-1}{\sqrt{\rho}-1} \cdot \frac{2L}{\lambda}\sum_{\phi \in A[i]} r_\phi\right).
\end{aligned}
$$

Now applying Theorem 3 using the Bernstein ellipse with parameter $\sqrt{\rho}$, we have, for any $v$ on the interval $[a, b]$,

$$\left| \tilde{f}(v) - \exp(\tilde{U}_v) \right| \leq \frac{4\rho^{-\frac{k}{2}}}{\sqrt{\rho} - 1} \cdot \exp\left( \frac{4\rho^{-\frac{m}{2}} + \sqrt{\rho} - 1}{\sqrt{\rho} - 1} \cdot \frac{2L}{\lambda} \sum_{\phi \in A[i]} r_\phi \right)$$

Therefore, it follows that

$$\left| \frac{\tilde{f}(v)}{\exp(U_v)} - 1 \right| \leq \left| \frac{\tilde{f}(v) - \exp(\tilde{U}_v) + \exp(\tilde{U}_v)}{\exp(U_v)} - 1 \right|$$

$$\leq \frac{\left| \tilde{f}(v) - \exp(\tilde{U}_v) \right|}{\exp(U_v)} + \left| \exp(\tilde{U}_v - U_v) - 1 \right|$$

$$\leq \left| \tilde{f}(v) - \exp(\tilde{U}_v) \right| + \exp\left( \left| \tilde{U}_v - U_v \right| \right) - 1,$$

where the last inequality is justified by the fact that $U_v$ is non-negative and for any $x$, $|\exp(x) - 1| \leq \exp(|x|) - 1$. Now substituting in our bounds from above gives us

$$\left| \frac{\tilde{f}(v)}{\exp(U_v)} - 1 \right|$$

$$\leq \exp\left( \frac{8\rho^{-\frac{m}{2}}}{\sqrt{\rho} - 1} \cdot \frac{L}{\lambda} \sum_{\phi \in A[i]} r_\phi \right) + \frac{4\rho^{-\frac{k}{2}}}{\sqrt{\rho} - 1} \cdot \exp\left( \frac{4\rho^{-\frac{m}{2}} + \sqrt{\rho} - 1}{\sqrt{\rho} - 1} \cdot \frac{2L}{\lambda} \sum_{\phi \in A[i]} r_\phi \right) - 1$$

As before, we let $B = \sum_{\phi \in A[i]} r_\phi$ where $B \sim \text{Poisson}(\Lambda)$. Then

$$\left| \frac{\tilde{f}(v)}{\exp(U_v)} - 1 \right| \leq \exp\left( \frac{8\rho^{-\frac{m}{2}}}{\sqrt{\rho} - 1} \cdot \frac{LB}{\lambda} \right) + \frac{4\rho^{-\frac{k}{2}}}{\sqrt{\rho} - 1} \cdot \exp\left( \frac{4\rho^{-\frac{m}{2}} + \sqrt{\rho} - 1}{\sqrt{\rho} - 1} \cdot \frac{2LB}{\lambda} \right) - 1$$

We define

$$E = \exp\left( \frac{8\rho^{-\frac{m}{2}}}{\sqrt{\rho} - 1} \cdot \frac{LB}{\lambda} \right) + \frac{4\rho^{-\frac{k}{2}}}{\sqrt{\rho} - 1} \cdot \exp\left( \frac{4\rho^{-\frac{m}{2}} + \sqrt{\rho} - 1}{\sqrt{\rho} - 1} \cdot \frac{2LB}{\lambda} \right) - 1,$$

and by following the same steps as used in Section C.1.1, with $E$ in place of the $C$ of that proof, we can get, for any constant $\delta > 0$,

$$\pi(x)T(x, y) \geq (1 - 2\delta) \cdot \exp\left( -\frac{4L^2}{\lambda} \right) \cdot \mathbf{P}_B(E \leq \delta)^2 \cdot \pi(x)G(x, y).$$

All that remains is to bound $\mathbf{P}_B(E \leq \delta)$. Using the MGF formula for $B$ twice, we get that

$$\mathbf{E}_B(E) = \frac{4\rho^{-\frac{k}{2}}}{\sqrt{\rho} - 1} \cdot \exp\left( \Lambda \left( \exp\left( \frac{4\rho^{-\frac{m}{2}} + \sqrt{\rho} - 1}{\sqrt{\rho} - 1} \cdot \frac{2L}{\lambda} \right) - 1 \right) \right)$$

$$+ \exp\left( \Lambda \left( \exp\left( \frac{8\rho^{-\frac{m}{2}}}{\sqrt{\rho} - 1} \cdot \frac{L}{\lambda} \right) - 1 \right) \right) - 1.$$

If we require that $m$ is large enough that

$$4\rho^{-\frac{m}{2}} \leq \sqrt{\rho} - 1,$$

then

$$\mathbf{E}_B(E) \leq \frac{4\rho^{-\frac{k}{2}}}{\sqrt{\rho} - 1} \cdot \exp\left( \Lambda \left( \exp\left( \frac{4L}{\lambda} \right) - 1 \right) \right)$$

$$+ \exp\left( \Lambda \left( \exp\left( \frac{8\rho^{-\frac{m}{2}}}{\sqrt{\rho} - 1} \cdot \frac{L}{\lambda} \right) - 1 \right) \right) - 1.$$

By Taylor's theorem, for $x > 0$,

$$\exp(x) - 1 = \exp(x) - \exp(0) \leq x \cdot \exp(x).$$

So, since $\Lambda \leq \lambda$, we can bound our expectation with

$$\mathbf{E}_B(E) \leq \frac{4\rho^{-\frac{k}{2}}}{\sqrt{\rho} - 1} \cdot \exp\left(\Lambda \cdot \frac{4L}{\lambda} \cdot \exp\left(\frac{4L}{\lambda}\right)\right)$$
$$+ \exp\left(\Lambda \cdot \frac{8\rho^{-\frac{m}{2}}}{\sqrt{\rho} - 1} \cdot \frac{L}{\lambda} \cdot \exp\left(\frac{8\rho^{-\frac{m}{2}}}{\sqrt{\rho} - 1} \cdot \frac{L}{\lambda}\right)\right) - 1$$
$$\leq \frac{4\rho^{-\frac{k}{2}}}{\sqrt{\rho} - 1} \cdot \exp\left(4L \cdot \exp\left(\frac{4L}{\lambda}\right)\right)$$
$$+ \exp\left(\frac{8\rho^{-\frac{m}{2}}}{\sqrt{\rho} - 1} \cdot L \cdot \exp\left(\frac{8\rho^{-\frac{m}{2}}}{\sqrt{\rho} - 1} \cdot \frac{L}{\lambda}\right)\right) - 1$$
$$\leq \frac{4\rho^{-\frac{k}{2}}}{\sqrt{\rho} - 1} \cdot \exp\left(4L \cdot \exp\left(\frac{4L}{\lambda}\right)\right)$$
$$+ \exp\left(\frac{8\rho^{-\frac{m}{2}}}{\sqrt{\rho} - 1} \cdot L \cdot \exp\left(\frac{4L}{\lambda}\right)\right) - 1.$$

Since $\lambda \log(2) \geq 4L$, we can bound $\exp(4L/\lambda) \leq 2$, and so

$$\mathbf{E}_B(E) \leq \frac{4\rho^{-\frac{k}{2}}}{\sqrt{\rho} - 1} \cdot \exp(8L) + \exp\left(\frac{16L\rho^{-\frac{m}{2}}}{\sqrt{\rho} - 1}\right) - 1.$$

We now define

$$F = \frac{4 \cdot \exp(8L) \cdot \rho^{-\frac{k}{2}}}{\sqrt{\rho} - 1} + \exp\left(\frac{16L\rho^{-\frac{m}{2}}}{\sqrt{\rho} - 1}\right) - 1.$$

By Markov's inequality,

$$\mathbf{P}_B(E \geq \delta) \geq \frac{\mathbf{E}_B(E)}{\delta} \geq F/\delta.$$

It follows

$$\mathbf{P}_B(E \leq \delta)^2 = (1 - \mathbf{P}_B(E \geq \delta))^2 \geq 1 - 2\mathbf{P}_B(E \geq \delta) \geq 1 - 2F/\delta.$$

Substituting it back into the overall bound,

$$\pi(x)T(x,y) \geq (1 - 2\delta) \cdot \exp\left(-\frac{4L^2}{\lambda}\right) \cdot \mathbf{P}_B(E \leq \delta)^2 \cdot \pi(x)G(x,y)$$
$$\geq \left(1 - 2\delta - \frac{2F}{\delta}\right) \cdot \exp\left(-\frac{4L^2}{\lambda}\right) \cdot \pi(x)G(x,y)$$

Let

$$\delta = \sqrt{F},$$

it becomes

$$\pi(x)T(x,y) \geq \left(1 - 4\sqrt{F}\right) \cdot \exp\left(-\frac{4L^2}{\lambda}\right) \cdot \pi(x)G(x,y)$$

Again, using the Dirichlet form we bound the spectral gap,

$$\bar{\gamma} \geq \left(1 - 4\sqrt{F}\right) \exp\left(\frac{-4L^2}{\lambda}\right) \cdot \gamma$$

$\square$

# D   Poisson-MH

We apply our Poisson-minibatching method to Metropolis-Hastings sampling. In Poisson-minibatching M-H (Poisson-MH), we first generate a candidate $x^*$ from the proposal distribution $q(x^*|x)$. Then the M-H ratio will be calculated as following

$$p = \frac{\exp\left(\sum_{\phi \in S} s_\phi \log\left(1 + \frac{L}{\lambda M_\phi}\phi(x^*)\right)\right) q(x^*|x)}{\exp\left(\sum_{\phi \in S} s_\phi \log\left(1 + \frac{L}{\lambda M_\phi}\phi(x)\right)\right) q(x|x^*)}$$

We accept $x^*$ with the probability $\min(1, p)$. After applying Poisson-minibatching, the M-H ratio no longer needs to use the whole dataset which will reduce the computational cost significantly.

Theorem 2 is similar to the bounds of Poisson-Gibbs. As long as we set $\lambda = \Theta(L^2)$, the convergence is slowed down by at most a constant factor which is unrelated to the size of the problem.

## D.1   Proof of Theorem 2

*Proof.* We begin with the transition probability from $x$ to $x^*$

$$
\begin{aligned}
& T(x^*, x) \\
&= \mathbf{E}\left\{q(x^*|x)\min\left(1, \frac{q(x|x^*)\pi(x^*, s)}{q(x^*|x)\pi(x, s)}\right)\right\} \\
&= \mathbf{E}\left\{q(x^*|x)\min\left(1, \frac{q(x|x^*)\exp\left(\sum_{\phi \in \Phi}\left[s_\phi \log\left(\frac{\lambda M_\phi}{L} + \phi(x^*)\right) - \log s_\phi!\right]\right)}{q(x^*|x)\exp\left(\sum_{\phi \in \Phi}\left[s_\phi \log\left(\frac{\lambda M_\phi}{L} + \phi(x)\right) - \log s_\phi!\right]\right)}\right)\right\} \\
&= \mathbf{E}\left\{q(x^*|x)\min\left(1, \frac{q(x|x^*)\exp\left(\sum_{\phi \in \Phi}\left[s_\phi \log\left(\frac{\lambda M_\phi}{L} + \phi(x^*)\right)\right]\right)}{q(x^*|x)\exp\left(\sum_{\phi \in \Phi}\left[s_\phi \log\left(\frac{\lambda M_\phi}{L} + \phi(x)\right)\right]\right)}\right)\right\} \\
&= \sum_s\left\{q(x^*|x)\min\left(1, \frac{q(x|x^*)\exp\left(\sum_{\phi \in \Phi}\left[s_\phi \log\left(\frac{\lambda M_\phi}{L} + \phi(x^*)\right)\right]\right)}{q(x^*|x)\exp\left(\sum_{\phi \in \Phi}\left[s_\phi \log\left(\frac{\lambda M_\phi}{L} + \phi(x)\right)\right]\right)}\right)\right\}\prod_{\phi \in \Phi}p(s_\phi|x) \\
&= \sum_s\left\{q(x^*|x)\min\left(\exp\left(\sum_{\phi \in \Phi}\left[s_\phi \log\left(\frac{\lambda M_\phi}{L} + \phi(x)\right) - \phi(x) - \frac{\lambda M_\phi}{L} - \log s_\phi!\right]\right),\right.\right. \\
&\qquad\qquad \left.\left.\frac{q(x|x^*)\exp\left(\sum_{\phi \in \Phi}\left[s_\phi \log\left(\frac{\lambda M_\phi}{L} + \phi(x^*)\right)\right]\right)}{q(x^*|x)\exp\left(\sum_{\phi \in \Phi}\phi(x) + \frac{\lambda M_\phi}{L} + \log s_\phi!\right)}\right)\right\} \\
&= \sum_s\left\{q(x^*|x)\min\left(\exp\left(\sum_{\phi \in \Phi}\left[s_\phi \log\left(\frac{\lambda M_\phi}{L} + \phi(x)\right) - \phi(x) - \frac{\lambda M_\phi}{L} - \log s_\phi!\right]\right),\right.\right. \\
&\qquad\qquad \left.\left.\frac{q(x|x^*)}{q(x^*|x)}\exp\left(\sum_{\phi \in \Phi}\left[s_\phi \log\left(\frac{\lambda M_\phi}{L} + \phi(x^*)\right) - \phi(x) - \frac{\lambda M_\phi}{L} - \log s_\phi!\right]\right)\right)\right\}
\end{aligned}
$$

Multiplying $\pi(x)$ to both sides,

$$\pi(x)T(x^*, x)$$

$$= \frac{1}{Z} \exp\left(\sum_{\phi \in \Phi} \phi(x)\right) T(x^*, x)$$

$$= \frac{1}{Z} \sum_s \min\left(q(x^*|x)\left(\exp\left(\sum_{\phi \in \Phi}\left[s_\phi \log\left(\frac{\lambda M_\phi}{L} + \phi(x)\right) - \frac{\lambda M_\phi}{L} - \log s_\phi!\right]\right)\right),\right.$$

$$\left. q(x|x^*)\exp\left(\sum_{\phi \in \Phi}\left[s_\phi \log\left(\frac{\lambda M_\phi}{L} + \phi(x^*)\right) - \frac{\lambda M_\phi}{L} - \log s_\phi!\right]\right)\right)\right)$$

This implies the Markov chain is reversible.

We can continue to reduce this to

$$\pi(x)T(x^*, x)$$

$$= \frac{1}{Z} \sum_s \min\left(q(x^*|x)\exp\left(\sum_{\phi \in \Phi} s_\phi \left[\log\left(\frac{\lambda M_\phi}{L} + \phi(x)\right) - \log\frac{\lambda M_\phi}{L}\right]\right),\right.$$

$$\left. q(x|x^*)\exp\left(\sum_{\phi \in \Phi} s_\phi \left[\log\left(\frac{\lambda M_\phi}{L} + \phi(x^*)\right) - \log\frac{\lambda M_\phi}{L}\right]\right)\right)$$

$$\cdot \prod_{\phi \in \Phi} \frac{1}{s_\phi!} \exp\left(-\frac{\lambda M_\phi}{L}\right)\left(\frac{\lambda M_\phi}{L}\right)^{s_\phi}$$

$$= \frac{1}{Z} \sum_s \min\left(q(x^*|x)\exp\left(\sum_{\phi \in \Phi} s_\phi \log\left(1 + \frac{L}{\lambda M_\phi}\phi(x)\right)\right),\right.$$

$$\left. q(x|x^*)\exp\left(\sum_{\phi \in \Phi} s_\phi \log\left(1 + \frac{L}{\lambda M_\phi}\phi(x^*)\right)\right)\right) \cdot \prod_{\phi \in \Phi} \frac{1}{s_\phi!} \exp\left(-\frac{\lambda M_\phi}{L}\right)\left(\frac{\lambda M_\phi}{L}\right)^{s_\phi}$$

Similar to the previous proof, $s_\phi$ here are non-negative integers that a Poisson variable can take, not variables. So if we let $r_\phi \sim \text{Poisson}\left(\frac{\lambda M_\phi}{L}\right)$ and $r_\phi$ to be all independent, we can write this as

$$\pi(x)T(x^*, x) = \frac{1}{Z}\mathbf{E}\min\left(q(x^*|x)\exp\left(\sum_{\phi \in \Phi} r_\phi \log\left(1 + \frac{L}{\lambda M_\phi}\phi(x)\right)\right),\right.$$

$$\left. q(x|x^*)\exp\left(\sum_{\phi \in \Phi} r_\phi \log\left(1 + \frac{L}{\lambda M_\phi}\phi(x^*)\right)\right)\right)$$

Assume $G(x^*, x)$ is the transition operator of a plain MCMC. Consider the ratio,

$$\frac{\pi(x)T(x^*, x)}{\pi(x)G(x^*, x)} = \frac{1}{Z}\mathbf{E}\min\left(q(x^*|x)\exp\left(\sum_{\phi \in \Phi} r_\phi \log\left(1 + \frac{L}{\lambda M_\phi}\phi(x)\right)\right),\right.$$

$$\left. q(x|x^*)\exp\left(\sum_{\phi \in \Phi} r_\phi \log\left(1 + \frac{L}{\lambda M_\phi}\phi(x^*)\right)\right)\right)$$

$$\cdot \left[1\Big/\left(\frac{1}{Z}\min\left(q(x^*|x)\exp\left(\sum_{\phi \in \Phi} \phi(x)\right), q(x|x^*)\exp\left(\sum_{\phi \in \Phi} \phi(x^*)\right)\right)\right)\right]$$

We know that $\frac{\min(A,B)}{\min(C,D)} = \min\left(\frac{A}{\min(C,D)}, \frac{B}{\min(C,D)}\right) \geq \min\left(\frac{A}{C}, \frac{B}{D}\right)$. The last inequality is due to the fact that $\frac{1}{\min(C,D)} \geq \frac{1}{C}$ and $\frac{1}{\min(C,D)} \geq \frac{1}{D}$.

With this inequality, we can continue simplifying the ratio,

$$\frac{\pi(x)T(x^*,x)}{\pi(x)G(x^*,x)} \geq \mathbf{E}\left[\min\left(\frac{\exp\left(\sum_{\phi\in\Phi} r_\phi \log\left(1 + \frac{L}{\lambda M_\phi}\phi(x)\right)\right)}{\exp\left(\sum_{\phi\in\Phi}\phi(x)\right)},\right.\right.$$
$$\left.\left.\frac{\exp\left(\sum_{\phi\in\Phi} r_\phi \log\left(1 + \frac{L}{\lambda M_\phi}\phi(x^*)\right)\right)}{\exp\left(\sum_{\phi\in\Phi}\phi(x^*)\right)}\right)\right]$$

$$= \mathbf{E}\left[\min\left(\exp\left(\sum_{\phi\in\Phi}\left(r_\phi\log\left(1+\frac{L}{\lambda M_\phi}\phi(x)\right) - \phi(x)\right)\right),\right.\right.$$
$$\left.\left.\exp\left(\sum_{\phi\in\Phi}\left(r_\phi\log\left(1+\frac{L}{\lambda M_\phi}\phi(x^*)\right) - \phi(x^*)\right)\right)\right)\right]$$

$$= \mathbf{E}\left[\max\left(\exp\left(\sum_{\phi\in\Phi}\left(\phi(x) - r_\phi\log\left(1+\frac{L}{\lambda M_\phi}\phi(x)\right)\right)\right),\right.\right.$$
$$\left.\left.\exp\left(\sum_{\phi\in\Phi}\left(\phi(x^*) - r_\phi\log\left(1+\frac{L}{\lambda M_\phi}\phi(x^*)\right)\right)\right)\right)^{-1}\right]$$

Because $f(x) = \frac{1}{x}$ is a convex function, by Jensen's inequality it follows

$$\frac{\pi(x)T(x^*,x)}{\pi(x)G(x^*,x)} \geq \mathbf{E}\left[\max\left(\exp\left(\sum_{\phi\in\Phi}\left(\phi(x) - r_\phi\log\left(1+\frac{L}{\lambda M_\phi}\phi(x)\right)\right)\right),\right.\right.$$
$$\left.\left.\exp\left(\sum_{\phi\in\Phi}\left(\phi(x^*) - r_\phi\log\left(1+\frac{L}{\lambda M_\phi}\phi(x^*)\right)\right)\right)\right)\right]^{-1}$$

We have that the maximum of the product is less than the product of maximum, therefore

$$\frac{\pi(x)T(x^*,x)}{\pi(x)G(x^*,x)} \geq \prod_{\phi\in\Phi}\mathbf{E}\left[\max\left(\exp\left(\phi(x) - r_\phi\log\left(1+\frac{L}{\lambda M_\phi}\phi(x)\right)\right),\right.\right.$$
$$\left.\left.\exp\left(\phi(x^*) - r_\phi\log\left(1+\frac{L}{\lambda M_\phi}\phi(x^*)\right)\right)\right)\right]^{-1}$$

Since $\max(A,B) \leq A + B$ when $A$ and $B$ are positive, it follows

$$\frac{\pi(x)T(x^*,x)}{\pi(x)G(x^*,x)} \geq \prod_{\phi\in\Phi}\mathbf{E}\left[\exp\left(\phi(x) - r_\phi\log\left(1+\frac{L}{\lambda M_\phi}\phi(x)\right)\right) +\right.$$
$$\left.\exp\left(\phi(x^*) - r_\phi\log\left(1+\frac{L}{\lambda M_\phi}\phi(x^*)\right)\right)\right]^{-1}$$

$\mathbf{E}\left[\exp\left(-r_\phi\log\left(1+\frac{L}{\lambda M_\phi}\phi(x)\right)\right)\right]$ is the moment generating function of the Poisson random variable $r_\phi$ evaluated at

$$t = -\log\left(1 + \frac{L}{\lambda M_\phi}\phi(x)\right)$$

We know that

$$\mathbf{E}\exp(r_\phi t) = \exp\left(\frac{\lambda M_\phi}{L}\left(\exp(t) - 1\right)\right)$$

Therefore,

$$\mathbf{E}\left[\exp\left(-r_\phi \log\left(1 + \frac{L}{\lambda M_\phi}\phi(x)\right)\right)\right] = \exp\left(-\frac{\phi(x)}{1 + \frac{L}{\lambda M_\phi}\phi(x)}\right)$$

Substituting this into the original expression produces

$$\frac{\pi(x)T(x^*, x)}{\pi(x)G(x^*, x)} \geq \left[2\prod_{\phi\in\Phi}\exp\left(-\frac{\phi(x)}{1 + \frac{L}{\lambda M_\phi}\phi(x)} + \phi(x)\right)\right]^{-1}$$

$$\geq \left[2\prod_{\phi\in\Phi}\exp(M_\phi)\exp\left(-\frac{1}{1 + \frac{L}{\lambda}} + 1\right)\right]^{-1}$$

$$= \left[2\prod_{\phi\in\Phi}\exp(M_\phi)\exp\left(\frac{L}{\lambda + L}\right)\right]^{-1}$$

$$= \left[2\exp\left(\frac{L^2}{\lambda + L}\right)\right]^{-1}$$

$$= \frac{1}{2}\exp\left(-\frac{L^2}{\lambda + L}\right)$$

From Dirichlet form argument, we get

$$\bar{\gamma} \geq \frac{1}{2}\exp\left(-\frac{L^2}{\lambda + L}\right)\cdot\gamma.$$

$\square$

### D.2 Additional Experiment: Poisson-MH on Truncated Gaussian Mixture

We test Poisson-MH on the truncated Gaussian mixture as in Section 4.3. The proposal is $q(x^*|x) = \mathcal{N}(x, 0.45^2 I)$. We set $\lambda = 500$. The estimated density is in Figure 4 which is very close to the true density. This demonstrates the effectiveness of Poisson-MH and the general applicability of Poisson-minibatching method.

Figure 4: The estimated density of Poisson-MH on a truncated Gaussian mixture model.

# E   Extended Results about Chebyshev Interpolants

In Trefethen [17], Theorem 8.2 proves bounds on the error of a Chebyshev interpolant on the interval $[-1, 1]$. However, in order to apply this theorem to a second Chebyshev interpolant that is a function of the first, we would need to bound the magnitude of that function *on a Bernstein ellipse*. To do this, we need the following extended version of Theorem 8.2, which bounds the error not only on the interval $[-1, 1]$ but more generally on a Bernstein ellipse.

**Lemma 1.** *Assume $U : \mathbb{C} \to \mathbb{C}$ is analytic in the open Bernstein ellipse $B([-1, 1], \rho)$, where the Bernstein ellipse is a region in the complex plane bounded by an ellipse with foci at $\pm 1$ and semimajor-plus-semiminor axis length $\rho > 1$. If for all $x \in B([-1, 1], \rho)$, $|U(x)| \leq V$ for some constant $V > 0$, then for any constant $1 < \sigma < \rho$, the error of the Chebyshev interpolant on the smaller Bernstein ellipse $B([-1, 1], \sigma)$ is bounded by*

$$|\tilde{U}(x) - U(x)| \leq \frac{4V}{\rho/\sigma - 1} \cdot \left(\frac{\rho}{\sigma}\right)^{-m}.$$

*Proof.* This proof is essentially identical to that of Theorem 8.2 in Trefethen [17], except that the error is bounded in a Bernstein ellipse rather than over only the real interval $[-1, 1]$.

First, note that one parameterization of the boundary of the Bernstein ellipse with parameter $\rho$ is

$$\left\{ \frac{z + z^{-1}}{2} \,\middle|\, z \in \mathbb{C},\ |z| = \rho \right\},$$

and the open ellipse itself can be written as

$$B([-1, 1], \rho) = \left\{ \frac{z + z^{-1}}{2} \,\middle|\, z \in \mathbb{C},\ \rho^{-1} \leq |z| \leq \rho \right\}.$$

Now, Theorem 8.1 from Trefethen [17] states that the Chebyshev coefficients of a function that satisfies the conditions of this theorem (boundedness and analyticity in a Bernstein ellipse) are bounded by $|a_0| \leq V$ and

$$|a_k| \leq 2V\rho^{-k},\ k \geq 1.$$

That is, for $a_k$ bounded in this way,

$$U(x) = \sum_{k=0}^{\infty} a_k T_k(x)$$

at least for all $x$ in the $\rho$-Bernstein ellipse on which $f$ is analytic. (While Trefethen [17] only states explicitly that this holds for $x \in [-1, 1]$, the fact that it also holds on the rest of the Bernstein ellipse follows directly from the fact that both sides of the equation are analytic over that region, using the identity theory for holomorphic functions.) Formula (4.9) from Trefethen [17] states that

$$U(x) - \tilde{U}_m(x) = \sum_{k=m+1}^{\infty} a_k \left(T_k(x) - T_{l(k,m)}(x)\right)$$

where $\tilde{U}_m$ denotes the degree-$m$ Chebyshev interpolant, and

$$l(k, m) = |((k + m - 1) \bmod 2m) - (m - 1)|.$$

Notice in particular that it always holds that $l(k, m) \leq m + 1$. Now, for $x$ inside the Bernstein ellipse $B([-1, 1], \sigma)$, there will always exist a $z \in \mathbb{C}$ such that $\sigma^{-1} \leq |z| \leq \sigma$ and

$$x = \frac{z + z^{-1}}{2}.$$

For such an $x$, and for any $k$,

$$|T_k(x)| = \left| T_k \left( \frac{z + z^{-1}}{2} \right) \right| = \left| \frac{z^k + z^{-k}}{2} \right| = \frac{|z|^k + |z|^{-k}}{2} \leq \sigma^k,$$

where the second equality is a well-known property of the Chebyshev polynomials. It follows that, for any $x$ in this Bernstein ellipse,

$$
\begin{aligned}
\left| U(x) - \tilde{U}_m(x) \right| &= \left| \sum_{k=m+1}^{\infty} a_k \left( T_k(x) - T_{l(k,m)}(x) \right) \right| \\
&\leq \sum_{k=m+1}^{\infty} |a_k| \cdot \left| T_k(x) - T_{l(k,m)}(x) \right| \\
&\leq \sum_{k=m+1}^{\infty} 2V\rho^{-k} \cdot \left( \sigma^k + \sigma^{l(k,m)} \right) \\
&\leq 4V \sum_{k=m+1}^{\infty} \rho^{-k}\sigma^k \\
&\leq 4V \left( \frac{\sigma}{\rho} \right)^{m+1} \sum_{k=0}^{\infty} \left( \frac{\sigma}{\rho} \right)^k \\
&\leq 4V \left( \frac{\sigma}{\rho} \right)^{m+1} \frac{1}{1 - \frac{\sigma}{\rho}} \\
&\leq 4V \left( \frac{\sigma}{\rho} \right)^{m} \frac{1}{\rho/\sigma - 1}.
\end{aligned}
$$

This is the desired result. $\qquad\qquad\square$