[Reviews · NeurIPS 2019]

Reviewer 1



Summary: This paper introduces Poisson auxiliary variables to facilitate minibatch sampling. The key insight is with the appropriate Poisson parameterization, the joint distribution (Eq. (1)) only depends on factors if they are in the minibatch. The authors apply this insight to discrete-state Gibbs sampling (Algorithm 2), Metropolis Hastings (Supplement), and continuous-state Gibbs sampling (Alg 3. and 5). The authors also develop spectral gap lower bounds for all proposed Gibbs sampling methods, which provides a rough guideline for choosing a tuning parameter $\lambda$ and comparing the (asymptotic) per iteration runtime of the methods (Table 1). Finally the authors evaluate the Gibbs methods on synthetic data, showing that their proposed method performs similarly to Gibbs while outperforming alternatives. Quality: The submission appears to be technically sound with only a few minor typos in the supplement. The computation speed-up claims are supported by the theoretical results (Theorems 1-5). For Theorem 5, I believe $\delta_k$ should be proportional to $exp(L + \delta_m)$ instead of $exp(L)$ as it requires an upperbound on $\tilde{U}$ rather than $U$. The computation costs in Table 1 seem to mix worst-case running time with average-case runtime (used to evaluate Poisson Gibbs); isn't the average-case per iteration runtime of plain Gibbs sampling O(D * mean degree) rather than O(D * max degree)? This is a minor complaint: in many applications (including the experiments of the paper), the mean degree = max degree = number of states. Originality: To the best of my knowledge, the idea to using Poisson-minibatching to reduce calculating factors is novel. The paper adequately cites previous work and distinguishes it contributions compared to previous related work of De Sa et al. (2018) in Table 1. Clarity: The paper is well written and clearly organized. The result on Poisson Minibatching for Metropolis-Hastings (although very interesting) seems a bit out of place, as it isn't introduced or addressed anywhere else in the main paper. In addition, there are a few areas that I suggest additional clarification: -Line 78: Algorithm 1 should be noted to only apply for discrete state spaces -As stated above, Table 1 and line 102 should be careful about whether computation cost per iteration is for the worst-case or average-case. -Line 112: "this joint distribution achieves the minibatch effect automatically", what is the minibatch effect? -Line 129: "At each iteration we will first re-sample all the $s_\phi$", but Algorithm 2 only samples $s_\phi$ for $\phi$ in $A[i]$. -Lines on Figure 1c are difficult to read printed. Significance: The Poisson Minibatching trick seems to be a useful contribution to the literature on scaling MCMC by subsampling factors. As demonstrated in the paper, there are many applications and extensions by combining this trick with existing sampling methods. The spectral gap convergences bounds are also nice and allow users to compare the trade-off between the different sampling schemes. Figure 1 of the experiments plots the performance of the proposed methods vs the number of Gibbs iterations. The computation speed-up claims would be better supported with the addition of identical plots with the x-axis changed to measured runtime. Typos: -Line 294: "the average needed steps to be accept is" -The last line in the equation immediately following line 373 should have $\exp(U_\nu(x))$ instead of $\exp(U(x))$ as the sum is over factors in $A[i]$ not all factors. -As a result, the next equation (immediately following line 374) appears to be missing a factor of $\exp(U(x)-U_\nu(x))$, but the result should still hold as this factor does not depend on $x[i]$. -In line 426, I believe it should read $m = \Theta(L)$ instead of $m = \Theta(L^2)$.

Reviewer 2



* Originality: The work builds off of previous work in a simple, but non-trivial way. The simplicity of the idea is appealing. Related work is adequately cited. * Quality: The proposed methods appear to be technically sound, and the bounds on the rates of convergence are useful. However, the experiments do little to shed light properties that the authors claim to be important. How does the compute/wall-clock time of the Poisson--Gibbs method compare to vanilla Gibbs and other mini-batched methods? How many potential functions are being evaluated at each iteration? Are there problems on which vanilla Gibbs would be prohibitively expensive for which Poisson--Gibbs would be useful (e.g., very large N for a highly connected graph)? Are there problems on which Poisson--Gibbs might fail (e.g., poor initialization and/or parameter settings; or strong dependence between variables in the graph)? These types of experiments are more useful than the current experiments. * Clarity: Overall, the main text of the paper is clearly written: the development is easy to follow, and the text proposes the main ideas in a simple manner that is consistent with the simplicity of those ideas. The text in the Sections 3 and 4 could use some cleaning/tightening. The proofs of the convergence rate theorems could be presented in a more reader friendly way, and should be checked for errors (for example, there appears to be a typo going from the equation after line 374 to that after line 378). * Significance: The ideas in the paper appear to be of moderate significance: the models to which the methods are applicable are somewhat limited, and no new technical or theoretical methods were introduced. * A few detailed comments on typos/technical points: - I did not review all of the proofs of the convergence rate theorems in detail, though on line 379 of the Supplemental Material (proof of Thm. 1) the authors suppose that the $s_{\phi}$ are iid Poisson random variables. Can the authors please explain why this supposition is ok/justified for the purposes of establishing the necessary bound? - The maximum local energy is a maximum taken over the entire set of variables, so 'local' seems a misnomer. I think the relevant property (as opposed to the total maximum energy is that it's not aggregated. - There's a stray ')' on line 82. Thanks to the authors for their thoughtful feedback to the initial reviews. In particular, the experiments showing wall clock and factor evaluation comparisons enhance the original submission. I have updated my review and score accordingly.

Reviewer 3



Update: I have gone over the comments from the other reviewers and the authors' response to the same. The wall-clock experiments provided by authors only reinforce my opinion that this paper should be clearly accepted. Originality: The paper seems to be a clever idea for speeding up minibatching methods, although at some level it seems natural to consider a Poisson auxilliary variable instead of the usual Bernoulli variables. Quality: The results in the paper are impressive considering usually methods based on minibatching has asymptotic biased. The authors here present clear results on the rates of convergence of the proposed methods, and provide reasonable simulations. Clarity: The article is fairly well written, and the concepts are clearly described. Significance: If minibatching is possible and useful in any problem, I would think that this particular article would be very pertinent to practitioners. However, it is unclear if this article (and any minibatching techniques) are universally useful for all problems.

[Author Response · NeurIPS 2019]

We thank all of the reviewers for their valuable comments and suggestions.

***Runtime Comparisons.*** As suggested by all reviewers, we add wall-clock time comparisons in order to demonstrate
empirically the computational speed-up of Poisson-Gibbs. We have replaced "Iterations" with "Time" in Figure 1
(using the same setup) and report the corresponding results in Figure 3. The number of factors being evaluated of
Poisson-Gibbs varies each iteration, so in Figure 3d, we report the average number of factors being evaluated per
iteration. Overall, the empirical results presented in Figure 3 align with our theoretical analysis: Poisson-Gibbs is
significantly faster than plain Gibbs on both the Potts model and the continuous spin model, and faster than other
minibatch Gibbs methods on the Potts (they cannot be applied to continuous domain). Compared to plain Gibbs,
Poisson-Gibbs speed up the computation by evaluating only a subset of factors in each iteration (Figure 3d). Compared
to DoubleMIN-Gibbs, Poisson-Gibbs is faster because it removes the need of an additional M-H correction step.

Figure 3: Runtime performance of Poisson-Gibbs, plain Gibbs and previous minibatch Gibbs methods on (a)-(b) a Potts model and (c) a continuous spin model. (d) Number of factors being evaluated per iteration.

***R2: Wall-clock time comparison and # of potential functions being evaluated.*** Please refer to Figure 3. Additionally,
the average number of factors being evaluated at each iteration in truncated Gaussian mixture is 1802 ($10^6$ in total).

***R2: Are there problems on which vanilla Gibbs would be prohibitively expensive?*** Yes, there are many applications
which use very large graphical models such as social and biological networks [12], which would make plain Gibbs
infeasible. However, in these cases our methods would be hard to compare empirically without the ground truth and the
results of plain Gibbs. This is why we focus on relatively large graphical models where it is still possible to run Gibbs
sampling, though with a fairly long runtime. We would be happy to investigate applying our methods to larger models
in the future work.

***R2: Are there problems on which Poisson-Gibbs might fail?*** Our theorems suggest that, as long as plain Gibbs can
converge well on this problem, our methods are guaranteed to perform similarly well by setting $\lambda$ as suggested in the
paper. The hyperparameters of the degree of Chebyshev polynomials in PGITS and PGDA need tuning for different
tasks (e.g. grid search). Empirically, we did not find a poor initialization issue for Poisson-Gibbs. One weakness of
Poisson-Gibbs is that it requires the domain to be bounded and the bound $L$ cannot be too large. If the problem has
a very large or even unbounded $L$, Poisson-Gibbs may not necessarily work efficiently. We would like to study the
problem of extending Poisson-Gibbs to unbounded domains in the future.

***R2: Why is the supposition that $s_\phi$ are iid ok?*** There is no additional supposition since the equation after line 379 is
just another way to write the equation after line 378. The equivalence is because $\sum_x x * p(x)$ (Eq 378) $= \mathbf{E}[x]$ (Eq 379).
Also, $s_\phi$ are independent but not identical since their parameters include different $M_\phi$. (We will clarify this.)

***R2: "local maximum energy" seems a misnomer.*** This name is from prior work [3], so for consistency we keep it.

***R3: Why not set $\lambda = 4L^2$ instead of $\lambda = L^2$ in simulation?*** When setting $\lambda$ larger (e.g. $4L^2$), the theoretical
convergence rate of Poisson-Gibbs becomes closer to plain Gibbs but the expected minibatch size upper bound $\lambda + L$
will become larger. So there is a trade-off. Also, setting $\lambda = 4L^2$ will result in $\exp(-1)$ in the bound, not canceling the
exponential. Empirically, we find that setting $\lambda = L^2$ gives reasonably good performance (see Figure 1b).

***R3: $\lambda = 500$ not $O(L^2)$ in truncated Gaussian mixture.*** We have considered higher values of $\lambda$ and find that setting
$\lambda = 500$ already provides good performance on this task. The theory guarantees that setting $\lambda = L^2$ can produce
similar performance as plain Gibbs. But in practice we can sometimes set $\lambda$ to be smaller to gain more speed-up.

***R1: $\delta_k$ should be proportional to $\exp(L + \delta_m)$ instead of $\exp(L)$ in Theorem 4.*** Yes, we have fixed this.

***R1: Clarification.*** For Table 1, we follow [3] to report the computational cost. For line 112, we want to say that our
disjoint distribution will only use a subset of factors which achieves the same effect as minibatching. For line 129, we
explain it in the later sentences (line 130-132) that we can do the sampling in a more efficient way and it is statistically
equivalent. We will clarify the above in the final version.

***Presentation.*** Thanks for all suggestions about the presentation. We will improve the presentation accordingly and
check proofs and typos carefully. We will present Poisson-MH in a better way and add a simple experiment for it.

[Meta-Review · NeurIPS 2019]

Congratulations on an elegant idea that, particularly with the addition of the experiments described in the rebuttal, could be of practical interest while also coming equipped with useful error bounds.